# GABAergic signaling linked to autophagy enhances host protection against intracellular bacterial infections

Jin Kyung Kim[1,2,3], Yi Sak Kim[1,2,3], Hye-Mi Lee[1,3], Hyo Sun Jin[4], Chiranjivi Neupane [2,5], Sup Kim[1,2,3], Sang-Hee Lee[6], Jung-Joon Min[7], Miwa Sasai[8], Jae-Ho Jeong [9,10], Seong-Kyu Choe[11], Jin-Man Kim[12], Masahiro Yamamoto[8], Hyon E. Choy [9,10], Jin Bong Park [2,5] & Eun-Kyeong Jo[1,2,3]

Gamma-aminobutyric acid (GABA) is the principal inhibitory neurotransmitter in the brain; however, the roles of GABA in antimicrobial host defenses are largely unknown. Here we demonstrate that GABAergic activation enhances antimicrobial responses against intracellular bacterial infection. Intracellular bacterial infection decreases GABA levels in vitro in macrophages and in vivo in sera. Treatment of macrophages with GABA or GABAergic drugs promotes autophagy activation, enhances phagosomal maturation and antimicrobial responses against mycobacterial infection. In macrophages, the GABAergic defense is mediated via macrophage type A GABA receptor (GABA$_A$R), intracellular calcium release, and the GABA type A receptor-associated protein-like 1 (GABARAPL1; an Atg8 homolog). Finally, GABAergic inhibition increases bacterial loads in mice and zebrafish in vivo, suggesting that the GABAergic defense plays an essential function in metazoan host defenses. Our study identified a previously unappreciated role for GABAergic signaling in linking antibacterial autophagy to enhance host innate defense against intracellular bacterial infection.

[1] Department of Microbiology, Chungnam National University School of Medicine, Daejeon 35015, Korea. [2] Department of Medical Science, Chungnam National University School of Medicine, Daejeon 35015, Korea. [3] Infection Control Convergence Research Center, Chungnam National University School of Medicine, Daejeon 35015, Korea. [4] Biomedical Research Institute, Chungnam National University Hospital, Daejeon 35015, Korea. [5] Department of Physiology, Chungnam National University School of Medicine, Daejeon 35015, Korea. [6] Institute of Molecular Biology & Genetics, Seoul National University, Seoul 08826, Korea. [7] Department of Nuclear Medicine, Chonnam National University Medical School, Gwangju 61469, Korea. [8] Department of Immunoparasitology, Research Institute for Microbial Diseases, Osaka University, Osaka 565-0871, Japan. [9] Department of Microbiology, Chonnam National University Medical School, Gwangju 61469, Korea. [10] Molecular Medicine, BK21 Plus, Chonnam National University Graduate School, Gwangju 61186, Korea. [11] Department of Microbiology and Center for Metabolic Function Regulation, Wonkwang University School of Medicine, Iksan, Jeonbuk 54538, Korea. [12] Department of Pathology, Chungnam National University School of Medicine, Daejeon 35015, Korea. These authors contributed equally: Jin Kyung Kim, Yi Sak Kim. Correspondence and requests for materials should be addressed to J.B.P. (email: jinbong@cnu.ac.kr) or to E.-K.J. (email: hayoungj@cnu.ac.kr)

Gamma-aminobutyric acid (GABA) is the major inhibitory neurotransmitter within the central nervous system and has been extensively studied in neurological disorders, such as epilepsy, anxiety disorders, and schizophrenia[1–3]. In neuronal cells, GABA is produced from glutamic acid by the action of glutamic acid decarboxylase (GAD) 65 and GAD67[3]. In turn, GABA is broken down by GABA transaminase to be recycled through the tricarboxylic acid cycle to form glutamic acid[3]. The optimal regulation of GABA-glutamic acid circuits is critical for the physiological function of the central nervous system, and its imbalance may trigger a myriad of symptoms associated with numerous neurological disorders[2,3]. Therefore, GABAergic medications that correct dysregulated GABAergic neurotransmission are being used to treat epilepsy, anxiety disorders, and schizophrenia, as well as in anesthesia[3]. A growing body of evidence has shown that a variety of peripheral immune cells express functional GABAergic components, including GABA-A ion channels, the GABA-B receptor, GABA transporters, and enzymes involved in GABA synthesis or degradation[4–6]. Preclinical studies suggest the potential application of GABA or GABAergic (agonistic) drugs as new therapeutics against autoimmune and inflammatory diseases, including type 1 diabetes, experimental autoimmune encephalomyelitis, and collagen-induced arthritis[7,8]. Nonetheless, given the potential role for GABA in the modulation of innate immune responses, it is poorly understood whether and how GABAergic signaling regulates antimicrobial host defenses in the context of bacterial infections.

During infection, the innate immune system regulates numerous host defense pathways, innate sensing, signal transduction, and the secretion of a variety of effector molecules, including antimicrobial proteins[9,10]. Autophagy is a well-recognized and conserved intracellular homeostatic mechanism through which damaged components or intracellular pathogens can be targeted and degraded via fusion with lysosomes[11,12]. As a crucial innate immune arm, the autophagy pathway provides a protective role against a variety of intracellular bacterial pathogens, including *Mycobacterium tuberculosis* (Mtb), *Salmonella enterica* serovar Typhimurium (*S. typhimurium*), and *Listeria monocytogenes* (*L. monocytogenes*)[13–16].

In this study, we showed that functional GABAergic system and GABA levels were modulated in innate immune cells upon intracellular bacterial infection. GABAergic activation led to increased antimicrobial responses against *Mycobacteria*, *Salmonella*, and *Listeria* infections in vitro and in vivo. We further showed that GABAergic defenses are mediated by host autophagy activation via the macrophage type A GABA receptor (GABA_AR), intracellular calcium increase, and AMP-activated protein kinase (AMPK) signaling. In addition, the autophagy gene GABA type A receptor-associated protein-like 1 (GABARAPL1), which belongs to the ATG8 family[17], is critically involved in the GABAergic induction of autophagy and phagosomal maturation against mycobacteria in macrophages. Furthermore, GABAergic inhibition increased bacterial loads and host susceptibility to intracellular bacterial infections in mice and zebrafish, suggesting that GABAergic signaling-associated autophagy plays a conserved function in metazoan host defense.

## Results

**Functional GABAergic system is modulated during infection.** Compared with extensive studies in brain and neural tissues, few studies have indicated that peripheral immune cells, such as macrophages and lymphocytes, contain molecular components of the GABAergic system[7,8,18,19]. To assess whether the GABAergic system in murine bone marrow-derived macrophages (BMDMs) is modulated during infection, we first used voltage-clamp

recording to confirm that BMDMs express functional GABA receptors (Fig. 1a). In the whole-cell patch-clamp mode, fast application of GABA to BMDMs using the Y-tube method evoked transient inward currents in a concentration-dependent manner (Fig. 1a). In the presence of bicuculline (BIC), a GABA_AR antagonist, GABA failed to cause an inward current ($n = 5$), suggesting that GABA predominantly evoked currents by binding to GABA_ARs. GABA-induced inward currents showed minimal desensitization during GABA application. Thus, we investigated whether GABA-induced tonic current, which was defined as a holding current shift by the GABA_AR antagonist BIC (50 μM), in the presence of GABA (Supplementary Fig. 1a). GABA generated tonic current in a concentration-dependent manner, which had similar amplitudes to those of the inward currents during the fast GABA application ($P > 0.5$ at each GABA concentration); this suggests that GABA mainly evoked tonic currents in BMDMs. GABA_AR pentamers, assembled from 19 subunits, vary in subunit combinations, which diversifies GABA_AR functions. For example, the δ-subunit confers a slower desensitization on GABA_ARs, making them ideal candidates to generate tonic GABA_A current in the brain[20]. To determine whether this is the case in peripheral macrophages, we used 4,5,6,7-tetrahydroisothiazolo-[5,4-c]pyridin-3-ol (THIP), which preferentially activates the δ over the γ_2 subunit-containing GABA_ARs[21–23] (Supplementary Fig. 1b). THIP (1–30 μM) generated tonic GABA_A current in a concentration-dependent manner in the presence of GABA (100 μM), although THIP alone failed to significantly alter the holding current of BMDMs. These results suggest that BMDMs express functional GABA_ARs that respond by generating tonic currents when exposed to GABA.

We next examined whether BMDMs and lung macrophages express GABA and how it is modulated by infection. As shown in Figs. 1b, c, endogenous GABA was stained in resting, unstimulated BMDMs, and this was decreased in Mtb-infected or *M. bovis* BCG (BCG)-infected BMDMs. GABA staining was primarily detected in uninfected BMDMs, whereas it was significantly decreased in Mtb- or BCG-infected BMDMs (Figs. 1b, c). In addition, GABA staining was observed in the lungs of Mtb-infected mice (Figs. 1d, e). GABA expression was highly colocalized with the macrophage marker F4/80, as well as the alveolar type II epithelial (AE II) cell-specific marker ABCA3 in murine lung tissues (Supplementary Fig. 2), suggesting that endogenous GABA is expressed by both lung AE II cells and macrophages in mouse lungs. Similar to BMDMs, GABA staining in the lungs was also decreased following infection (Fig. 1e). Interestingly, the secreted GABA levels were significantly decreased in sera from Mtb-infected mice (Fig. 1f). The specificity of anti-GABA antibody was confirmed by isotype control antibody and secondary antibody controls (Supplementary Fig. 3a, b).

We also found that GAD65/67 is expressed in resting and uninfected BMDMs, and that these levels were increased upon Mtb or BCG infection (Figs. 1g, h and Supplementary Fig. 4a–d). Furthermore, we performed real-time quantitative reverse transcription polymerase chain reaction (qRT-PCR) analysis for a total of 19 subunits of GABA_AR[24]. As shown in Fig. 1i, the mRNAs of 11 (*Gabra2*, *Gabra3*, *Gabra4*, *Gabra5*, *Gabra6*, *Gabrb2*, *Gabrb3*, *Gabrg1*, *Gabrg2*, *Gabrd*, and *Gabrq*) out of 19 GABA_AR subunits were expressed, whereas the mRNA levels of the other subunits were undetectable in resting, unstimulated BMDMs. We then selected five subunits (*Gabra3*, *Gabra4*, *Gabrb3*, *Gabrd*, and *Gabrq*) that were detectable in BMDMs, and examined their mRNA levels before and after Mtb infection. Interestingly, all of them were downregulated in BMDMs after infection in a time-dependent manner (Fig. 1j). Similarly, the mRNA levels of the GABA_AR subunits *Gabra3*, *Gabra4*, *Gabrb3*,

and *Gabrd* were decreased in the lung tissues of Mtb-infected mice (Fig. 1k). Together, these data demonstrate that the components of the GABAergic system are present in BMDMs, and that the levels of GABA and GABA$_A$R subunits are modulated by mycobacterial infection.

**GABAergic activation promotes host protective responses.** To explore whether GABA or GABAergic activation amplified the host defense against microbial infection, C57BL/6 mice were infected with Mtb or BCG, and then either treated with GABA, or GABAergic activation of the GABA$_A$R was triggered by injecting mice with muscimol (Mus), a GABA structural analog. Relative to the vehicle control, intraperitoneal (i.p.) injection of either GABA or Mus significantly reduced the Mtb and BCG bacterial burdens in various tissues (Figs. 2a–c and Supplementary Fig. 6a–c). Additionally, the number of granulomatous lung lesions and neutrophil infiltration decreased significantly (compared with controls) following GABA treatment after intravenous (i.v.) Mtb

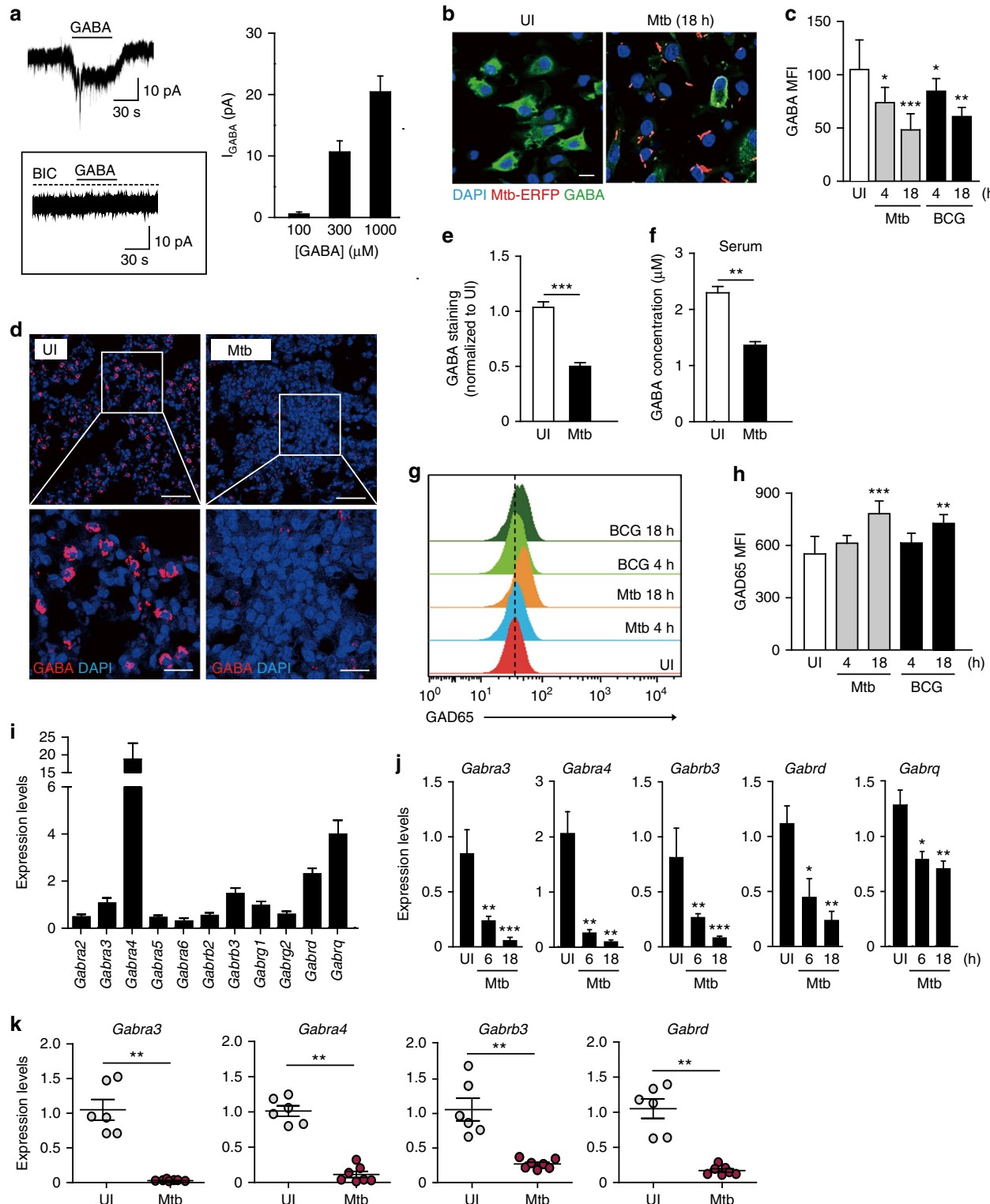

**Fig. 1** The macrophage GABAergic system and GABA levels during infection. **a** Representative GABA-induced whole-cell currents in BMDMs (GABA, 300 μM). Bottom, the effects by $GABA_AR$ antagonist bicuculline (BIC, 50 μM). Right, quantification of GABA-induced currents at each GABA concentration. **b**, **c** BMDMs were infected with Mtb-ERFP (MOI of 5) for the indicated times and stained with GABA (green) and DAPI (nuclei; blue). **b** Cells were visualized by confocal microscopy. Scale bars, 15 μm. **c** Average mean fluorescence intensities (MFIs) of GABA. **d**–**f** Mice were infected i.v. with Mtb ($1 \times 10^6$ CFU), and monitored at 14 dpi. Lung tissues were stained with GABA (red) and DAPI (nuclei; blue). **d** Representative in vivo imaging data from multiple lesions of Mtb-infected or -uninfected lungs ($n = 8$ per group). Scale bars, 50 μm (top) and 20 μm (bottom). **e** Average MFIs of GABA staining ($n = 4$ per group). **f** GABA in serum was measured by ELISA ($n = 8$ per group). **g**, **h** BMDMs were infected with Mtb (MOI of 5) or BCG (MOI of 5) for the indicated times. **g** GAD65 expression analyzed by flow cytometry. The units of the horizontal axis are the fluorescence (FITC) intensity. Representative gating strategy is shown in Supplementary Fig. 5. **h** Average MFIs of GAD65. **i** In BMDMs, 11 $GABA_AR$ subunit mRNAs were determined using the ΔΔCt method, and normalized to the reference gene *Gapdh*. **j** BMDMs were infected with Mtb (MOI of 5) for the indicated times and $GABA_AR$ subunit expression was determined using qRT-PCR. **k** $GABA_AR$ subunit expression in lung tissues was determined using qRT-PCR ($n = 6$ mice for no infection, $n = 7$ mice for Mtb infection) as in **d**–**f**. *$p < 0.05$, **$p < 0.01$, and ***$p < 0.001$. ns: not significant, UI: uninfected. Statistical significance was determined by one-way ANOVA (**c**, **h**, **j**), Mann–Whitney $U$-test (**f**, **k**), or two-tailed unpaired $t$-test (**e**). Data shown (means ± SEM) represent the combined results of triplicates from three independent experiments (**c**, **h**, **i**, **j**), or of 15 (**a**), 2 (**f**), or 3 (**b**–**e**, **g**) independent experiments. Each symbol represents one animal (**k**)

(Fig. 2d, for granuloma; Supplementary Fig. 6d, for neutrophil infiltration) or i.v. BCG challenge (Supplementary Fig. 6b). We then used *M. marinum* to establish a zebrafish embryo model of infection. Infected embryos were exposed to GABA and bacterial counts were measured. As shown in Supplementary Fig. 7, the bacterial burdens were significantly lower in zebrafish embryos grown with GABA. Consistent with the in vivo data, the intracellular survival rates of Mtb in both murine BMDMs (Fig. 2e) and human monocyte-derived macrophages (MDMs) (Fig. 2f) were significantly reduced upon treatment with GABA or $GABA_AR$ agonists (Mus or isoguvacine hydrochloride [Iso]) in a dose-dependent manner. In human primary MDMs, we detected three mRNAs (*GABRA1*, *GABRB1*, and *GABRR2*) by qRT-PCR analysis (Supplementary Fig. 8).

We further examined whether GABA treatment led to significant inhibition of tumor necrosis factor (TNF) and interleukin 6 (IL6) in lung tissues and BMDMs during Mtb infection. We found that *Tnf* and *Il6* mRNA levels were significantly decreased in the infected lung samples by GABA treatment, when compared with those from untreated/Mtb-infected controls (Supplementary Fig. 9a, b). In BMDMs, GABA treatment led to a significant reduction in TNF-α and IL-6 protein expression in response to Mtb infection (Supplementary Fig. 9c, d). Taken together, these results indicate that GABA treatment inhibited proinflammatory cytokine production in vivo and in vitro during Mtb infection.

We explored whether GABA enhanced the host defense against another intracellular bacterial infection. We infected C57BL/6 mice with *S. typhimurium* and administered GABA by i.p. injection. Pre- and post-treatment of mice with GABA resulted in a reduced susceptibility when challenged with *S. typhimurium* (Figs. 2g, h). We also observed enhanced animal survival in *S. typhimurium*-infected mice (Fig. 2g) and decreased *S. typhimurium* viability in the liver, spleen, and mesenteric lymph node (MLN) (Fig. 2h) after i.p. administration of GABA. In addition, we examined whether GABA treatment played a critical role for innate immune defense in macrophages. When clodronate liposomes were administered, the GABAergic protective effects upon mouse survival in *S. typhimurium* infection were completely abrogated (Supplementary Fig. 10a). In addition, clodronate-liposome treatment resulted in equal bacterial growth in both GABA-treated and untreated control mice (Supplementary Fig. 10b and 10c, for *S. typhimurinum* and BCG, respectively). Together, these results clearly show that GABAergic activation triggered the antimicrobial host defense and ameliorated pathologic inflammation against intracellular bacterial infections. In addition, macrophages may play a critical role in GABAergic protective responses against intracellular bacterial infection.

**GABAergic activation enhances autophagy in macrophages**. To define the mechanism by which GABA triggers the antimicrobial host defense system during intracellular bacterial infection, we used RNA sequencing (RNAseq) to analyze the GABA-mediated, genome-wide transcriptional changes in BMDMs. GABA treatment evoked coordinated differences in the expression of genes involved in macrophage effector functions, including autophagy-related genes (ATGs; *Gabarapl1*, *Gabarapl2*, *Map1lc3b*, *Atg16l2*, and *Lamp1*); genes of Toll-like receptor (TLR) signaling (*Cd180*, *Tlr4*, *Ticam2*, *Tirap*, *Irak2*, and *Traf6*); and genes involved in T-cell responses (*Ccr5*, *Il12rb2*, and *Il18r1*). In particular, numerous genes involved in autophagy were substantially upregulated in GABA-treated cells (Fig. 3a). Given that GABA induced the transcriptional activation of ATGs, we next explored whether GABA activated autophagy. Thus, we examined the expression levels of mRNAs encoding genes associated with autophagic induction and maturation, as well as autophagosome formation. The qRT-PCR analysis showed that GABA significantly increased the mRNA expression levels of 8 of 12 genes involved in autophagosome formation and autolysosome maturation[25] in BMDMs (Fig. 3b). We further showed that the numbers of LC3-positive autophagic punctae and LC3B protein expression levels increased significantly after either GABA addition or GABAergic activation (via Mus or Iso) (Figs. 3c–e and Supplementary Fig. 11a, b). Western blot analysis further showed that treatment of BMDMs with GABA or $GABA_AR$ agonists robustly increased the autophagosomal membrane-associated LC3-II fractions (Fig. 3f). In addition, GABA-induced LC3-II fractions were further accumulated by pretreatment with the vacuolar $H^+$-ATPase inhibitor, bafilomycin A1, indicating a real autophagic flux in BMDMs (Fig. 3f). Ultrastructurally, GABA increased the numbers of autophagic vesicles (autophagosomes/autolysosomes) in BMDMs (Figs. 3g, h). We used a tandem-LC3 retroviral vector (mCherry-EGFP-LC3B) to confirm that GABA triggered genuine autophagic flux, which is associated with increased autolysosome numbers, in BMDMs (Figs. 3i, j). These data reveal that either exogenous GABA treatment or GABAergic activation triggers the activation of autophagy in macrophages.

**GABAergic autophagy promotes Mtb phagosomal maturation**. Apart from playing a fundamental role in the maintenance of intracellular homeostasis[13], autophagy both targets and protects against a variety of intracellular bacterial pathogens, including Mtb, *S. typhimurium*, and *L. monocytogenes*[15,16]. We thus explored whether GABA-induced autophagy activation was required for phagosomal maturation during Mtb infection. As shown in Figs. 4a, b, the formation of LC3 punctae and the extent of their colocalization with bacterial phagosomes were greatly

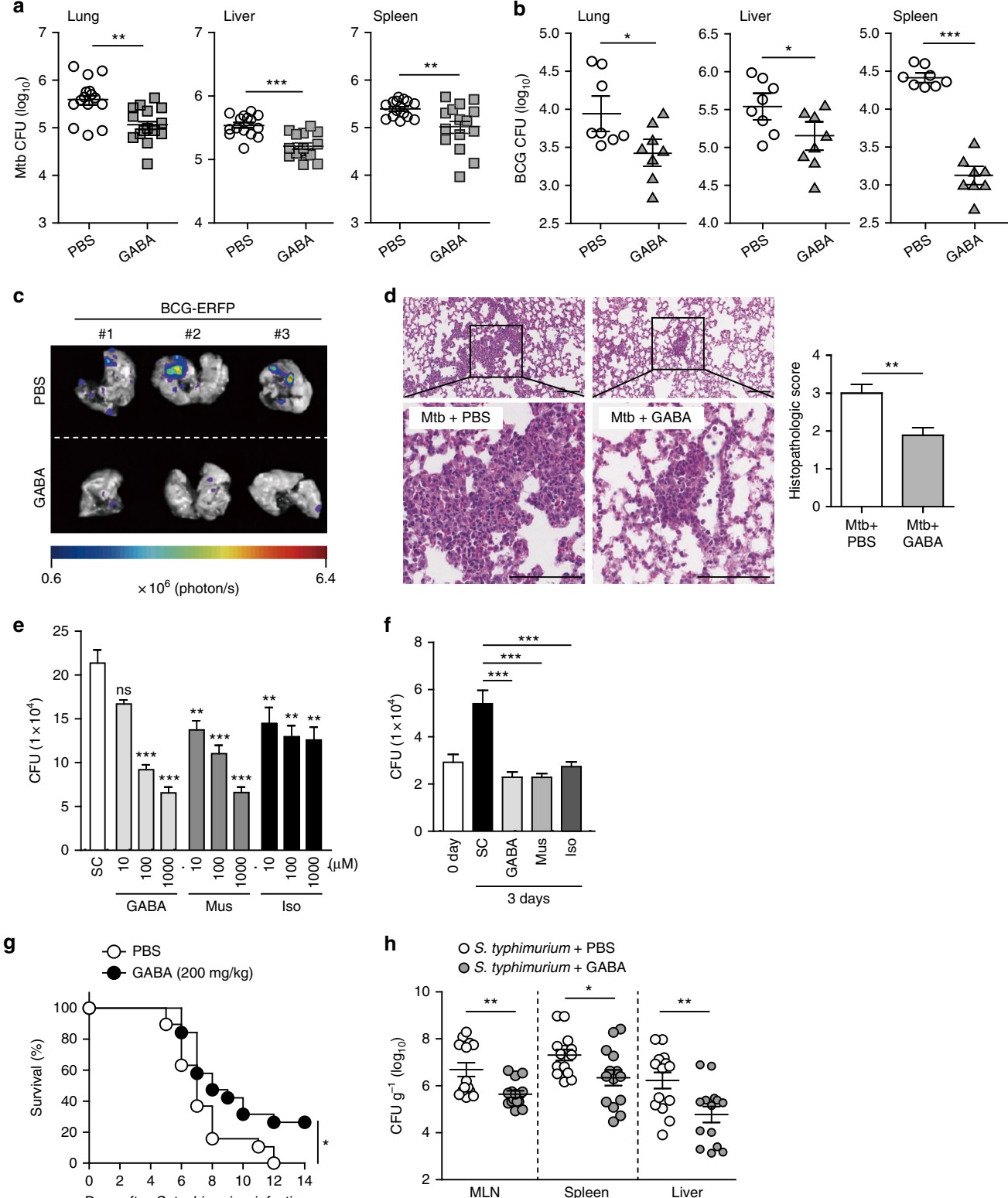

increased in Mtb-infected/GABA-treated BMDMs compared with Mtb-infected/untreated BMDMs. In addition, lysosome numbers and the extent of their colocalization with mycobacterial phagosomes were significantly enhanced in the former cells (Figs. 4c, d). Furthermore, GABAergic activation by GABA$_A$R agonists (Mus or Iso) significantly increased the proportion of Mtb within both autophagosomal (Figs. 4a, b) and lysosomal

(Figs. 4c, d) structures. Thus, both GABA and GABAergic activation enhanced phagosomal maturation during Mtb infection.

The effects of GABAergic activation on xenophagic fluxes against Mtb and BCG were further explored in infected lungs in vivo. Importantly, ultrastructural analysis revealed that GABA treatment enhanced bacterial numbers within autophagic vesicles in lungs and in macrophages (Figs. 4e, f for lungs and

**Fig. 2** GABAergic activation promotes antimicrobial responses against mycobacterial and salmonella infection. **a, b** In vivo bacterial loads determined by CFU assay. **a** Mice ($n = 15$ per group) were infected i.v. with Mtb ($1 \times 10^6$ CFU), treated with PBS or GABA (daily i.p. 200 mg/kg), and monitored at 14 dpi. **b** Mice ($n = 8$ per group) were infected i.v. with BCG ($1 \times 10^7$ CFU), treated with PBS or GABA (daily i.p. 200 mg/kg), and monitored at 10 dpi. **c** Representative in vivo imaging of BCG-ERFP-infected lungs (i.n., $2 \times 10^6$ CFU from mice treated with PBS or GABA (daily i.p. 200 mg/kg $n = 3$ per group) at 7 dpi. **d** Representative H&E-stained images (left) in lung tissue of mice treated as in **a**, and quantitative analysis of histopathology scores (right). Scale bars, 100 μm. **e** Intracellular survival of Mtb assessed in BMDMs treated with GABA (10, 100, or 1000 μM), muscimol (Mus; 10, 100, or 1000 μM), or isoguvacine hydrochloride (Iso; 10, 100, or 1000 μM) for 3 days. **f** Human MDMs were infected with Mtb (MOI of 5), followed by treatment with GABA (100 μM), Mus (100 μM), or Iso (100 μM). Intracellular survival of Mtb was determined by CFU assay. **g, h** After *S. typhimurium* ($1 \times 10^8$ CFU) oral infection, PBS or GABA (200 mg/kg) were injected daily i.p. for 7 days. **g** Survival of mice ($n = 18$–19 per group). **h** Viable cell count of intracellular *S. typhimurium* in MLN, spleen, and liver of mice treated with PBS or GABA ($n = 14$ per group) at 5 dpi. *$p < 0.05$, **$p < 0.01$, and ***$p < 0.001$. ns: not significant, SC: solvent control (0.1% PBS). Statistical significance was determined by unpaired *t*-test (**a, b, h**), Mann–Whitney *U*-test (**d**), one-way ANOVA (**e, f**), and log-rank (Mantel–Cox) test (**g**). Data shown (means ± SEM) represent the combined results of two (**a**) or three (**b, d–h**) independent experiments. Images are representative of one (**c**) or three (**d**) independent experiments. Each symbol represents one animal (**a, b, h**)

Supplementary Fig. 12a, b for BMDMs). In addition, GABA significantly reduced the level of p62 in the lung tissues of mice infected with Mtb 2 weeks later (Figs. 4g, h), suggesting that GABA upregulated the Mtb xenophagic flux in vivo. To further examine the role of GABAergic autophagy in the antimicrobial responses mounted during mycobacterial infection, BMDMs from *Atg7*^fl/fl^LysM-Cre^+^ (*Atg7* knockout [KO]) mice (test) and their littermates (*Atg7*^fl/fl^LysM-Cre^-^) (control) were infected with Mtb, followed by the addition of GABA. The enhancement of GABAergic Mtb killing evident in *Atg7* control BMDMs was significantly abrogated in *Atg7*-KO BMDMs (Fig. 4i). Therefore, GABAergic activation of autophagy is required for phagosomal maturation and the development of antimicrobial responses during mycobacterial infection.

**GABAergic autophagy depends on GABA$_A$R-Ca$^{2+}$-AMPK signaling.** We further explored the mechanism by which GABA activates both autophagy and phagosomal maturation in vitro and in vivo. As GABAergic activation triggered host defenses, we examined the effects of GABA$_A$R antagonists (picrotoxin [PTX] or BIC) on GABA-induced autophagy and the maturation of phagosomes containing mycobacteria. Treatment of BMDMs with the GABA$_A$R antagonists significantly abolished GABA-induced LC3 autophagosome formation and LC3B expression (Figs. 5a, b and Supplementary Fig. 13a, b). Further, blockade of GABA$_A$R by pharmacological levels of PTX or BIC significantly inhibited the GABA-mediated increase in the colocalization of Mtb phagosomes with autophagosomes (Figs. 5c, d) and lysosomes (Supplementary Fig. 13c, d).

Earlier studies found that the GABA transport system was modulated by incoming cationic fluxes, including a Ca$^{2+}$ flux[26]. In addition, AMPK signaling is required for antibacterial autophagy against Mtb[27,28]. We thus explored whether altered intracellular Ca$^{2+}$ concentration ([Ca$^{2+}$]$_i$) or AMPK activation is required for the GABAergic activation of autophagy and phagosomal maturation against Mtb. Treatment of BMDMs with GABA rapidly increased [Ca$^{2+}$]$_i$ (Fig. 5e, top) and phosphorylation of AMPK (Fig. 5f). Blockade of [Ca$^{2+}$]$_i$ increase (by pretreatment with 1,2-bis-(o-aminophenoxy)-ethane-*N,N,N′,N′*-tetraacetic acid, tetraacetoxymethyl ester (BAPTA-AM), the chelator of intracellular calcium; Supplementary Fig. 14) or silencing of *Ampk* (by a lentiviral short hairpin RNA [shRNA] specific to *Ampk*) significantly reduced the extent of both autophagic activation (Figs. 5g, h and Supplementary Fig. 13e, f) and phagosomal maturation during Mtb infection (Figs. 5i, j). It was also noted that pre-incubation of a selective GABA$_A$R antagonist, BIC, efficiently blocked [Ca$^{2+}$]$_i$ increase induced by GABA but not by ATP (Fig. 5e, bottom), indicating that GABA-induced intracellular calcium release is mediated through

GABA$_A$R activation. In addition, GABA-induced AMPK activation was significantly inhibited in BMDMs in the presence of either BIC or BAPTA-AM in a dose-dependent manner (Supplementary Fig. 15a, b), suggesting the contribution of GABA$_A$R or [Ca$^{2+}$]$_i$ increase in GABA-mediated activation of AMPK. Further, the silencing of *Ampk* significantly inhibited the GABAergic transcriptional activation of several ATGs, including *Gabarapl1* mRNA expression, in BMDMs (Fig. 5k). These data suggest that GABA$_A$R, [Ca$^{2+}$]$_i$ increase, and AMPK signaling contribute to the GABAergic activation of autophagy and phagosomal maturation in macrophages.

**GABARAPL1 is essential for antimicrobial responses to Mtb.** GABARAPL1 is a key autophagy-associated protein important for autophagosomal maturation and lysosomal activity[29–31], and is involved in the anti-toxoplasmal response[32]. Of the ATG candidates explored via RNAseq analysis, the level of mRNA encoding the LC3-associated protein *Gabarapl1* exhibited the greatest increase among the ATG-encoding mRNAs following GABA treatment (Fig. 3b). We thus examined the effects of GABA on the expression of LC3-associated proteins of the GABARAPL1 family during infection. Intriguingly, GABAergic activation increased the levels of mRNAs encoding *Gabarapl1*, but not *Gabarapl2*, in Mtb-infected BMDMs (Fig. 6a).

Using GABARAPL1-overexpressing RAW264.7 cells, we showed that GABARAPL1 strongly facilitated the colocalization of bacteria and the lysosomal protein LAMP2 (Figs. 6b, c), as well as antimicrobial responses against Mtb infection (Fig. 6d). In RAW264.7 cells, we found that the mRNAs of several GABA$_A$R subunits (*Gabra2*, *Gabra3*, *Gabra5*, *Gabrb3*, and *Gabrd*) were present, as measured by qRT-PCR analysis (Supplementary Fig. 16). We then examined whether GABARAPL1 is required for GABA-induced phagosomal maturation against mycobacteria using *Gabarapl1*-null BMDMs[32]. In *Gabarapl1*-null BMDMs, the GABA-mediated colocalization of bacterial phagosomes with lysosomes was dramatically decreased (Supplementary Fig. 17a, b), suggesting that GABARAPL1 is required for the phagosomal maturation of Mtb. Additionally, *Gabarapl1* silencing markedly reduced the colocalization of bacterial phagosomes with autophagosomes (Supplementary Fig. 17c, d) or lysosomes (Figs. 6e, f), confirming that GABARAPL1 plays an essential role in the phagosomal maturation of Mtb. Importantly, the antimycobacterial activities induced by GABA were significantly attenuated in BMDMs after *Gabarapl1* knockdown (Fig. 6g) or in *Gabarapl1*-null BMDMs (Fig. 6h). The GABAergic signaling linked to autophagy for host protection against intracellular bacterial infections is summarized in Supplementary Fig. 17e. These data strongly suggest that GABARAPL1 is required for GABA-

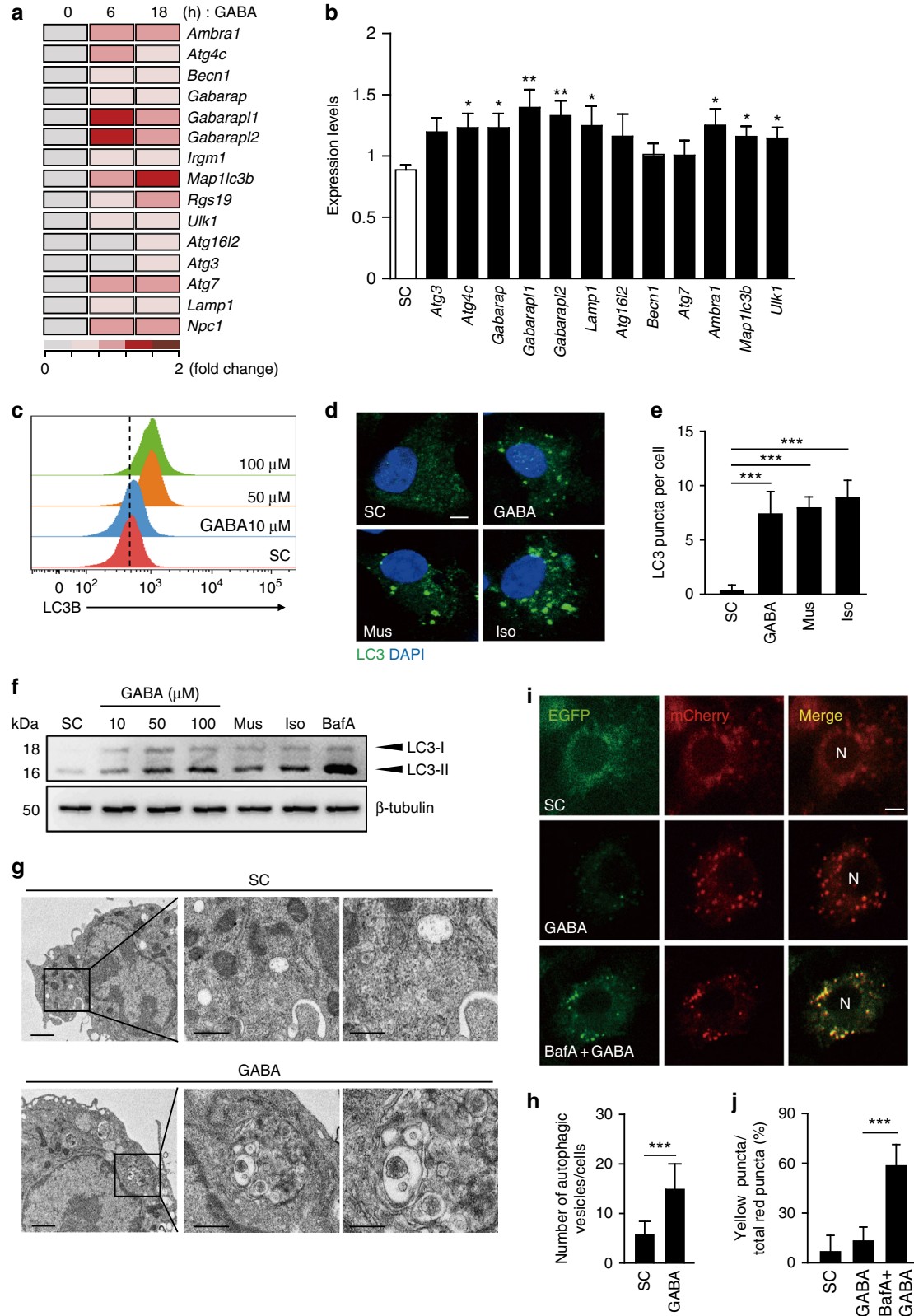

induced phagosomal maturation and antimicrobial responses against mycobacterial infection.

**GABAergic inhibition decreases host protective responses.** These data collectively show that both GABA synthesis per se and GABAergic activation are essential when host defenses are mounted against mycobacterial infections. We thus hypothesized that systemic GABAergic inhibition would increase host susceptibility to such infection. Mice were pretreated with low-dose (20 or 40 mg/kg) pentylenetetrazole (PTZ; a blocker of the GABA$_A$R-associated Cl$^-$ channel) 6 h or 7 days before infection (such doses do not trigger epileptic seizures)[33,34]. Next, mice were infected with Mtb or BCG. PTZ significantly enhanced

**Fig. 3** GABAergic activation triggers autophagy in macrophages. **a** Heatmap analysis of the RNAseq data (representative of duplicate determinations). BMDMs were treated with GABA (100 μM) for the indicated times and then subjected to RNAseq analysis. **b** BMDMs were treated with GABA (100 μM) for 6 h. Cells were subjected to qRT-PCR to determine mRNA levels of autophagy genes. **c** BMDMs were treated with GABA (10, 50, or 100 μM) for 18 h and the expression of LC3B was analyzed by flow cytometry (horizontal axis, fluorescence (FITC) intensity). Representative gating strategy is shown in Supplementary Fig. 5. **d**, **e** BMDMs were incubated with GABA (100 μM), muscimol (Mus; 100 μM), or isoguvacine hydrochloride (Iso; 100 μM) and then stained with LC3 (green) and DAPI (nuclei; blue). **d** Confocal microscopic analysis. Scale bars, 5 μm. **e** Quantitative data of LC3 punctate analysis. **f** BMDMs were pretreated with or without bafilomycin A1 (BafA, 200 nM; 2 h) and incubated with GABA (10, 50, or 100 μM), Mus (100 μM), or Iso (100 μM) for 18 h. LC3 and β-tubulin levels were evaluated by immunoblotting. **g**, **h** BMDMs were incubated with GABA (100 μM) for 24 h. **g** Representative transmission electron micrograph of BMDMs. Scale bars, 1 μm (left), 500 nm (middle), and 200 nm (right). **h** Quantitative data of autophagic vesicle numbers. **i**, **j** BMDMs were transduced with a retrovirus expressing a tandem LC3B plasmid (mCherry-EGFP-LC3B). Cells were pretreated without or with BafA (200 nM; 2 h) and incubated with GABA (100 μM) for 18 h. **i** Confocal microscopic analysis for LC3. Scale bars: 5 μm. **j** Quantification of yellow puncta/total red puncta (%) per cell. *$p < 0.05$, **$p < 0.01$, and ***$p < 0.001$. U: untreated, SC: solvent control (0.1% PBS, for **c–e**, **g**, **h**; 0.1% dimethyl sulfoxide, for **f**, **i**, **j**), N: nucleus. Statistical significance was determined by Mann–Whitney U-test (**b**), one-way ANOVA (**e**, **j**), or unpaired t-test (**h**). Data shown (means ± SEM) represent the combined results of triplicates from three independent experiments (**b**, **e**, **h**, **j**). Images are representative of three or four (**a**, **c**, **d**, **f**, **g**, **i**) independent experiments

mycobacterial growth and the number of lung granulomatous lesions (Figs. 7a–d and Supplementary Fig. 18a). Mycobacterial growth in BMDMs was also markedly increased upon treatment with the GABA$_A$R antagonist PTX 3 days after infection (Fig. 7e).

We next used *M. marinum* to establish a zebrafish embryo model of infection. Infected embryos were exposed to PTX, BIC, or PTZ (GABA$_A$R antagonists), and bacterial counts and survival levels were measured. As shown in Figs. 7f, g, imaging and fluorescent pixel counting showed that the bacterial burdens were higher in zebrafish embryos growing with PTX, BIC, or PTZ. Additionally, such treatments increased the severity of infection and rendered the embryos hypersusceptible to *M. marinum*, as shown in Fig. 7h. In addition, in vivo phagosomal maturation, as assessed by the extent of colocalization of *M. marinum* and lysotracker dye, was markedly decreased by BIC or PTX in infected zebrafishes (Supplementary Fig. 18b, c). Furthermore, GABAergic inhibition of W$^{1118}$ flies significantly increased the susceptibility to *M. marinum* (Supplementary Fig. 18d).

We next assessed whether GABA$_A$R antagonists affected the host defense against *S. typhimurium* or *L. monocytogenes* infection. Pretreatment of mice with the GABA$_A$R antagonist PTZ greatly increased the susceptibility to *S. typhimurium* or *L. monocytogenes* infection (Figs. 7i, j and Supplementary Fig. 18e, f). The in vivo *S. typhimurium* or *L. monocytogenes* bacterial load increased markedly in various tissues (Fig. 7i and Supplementary Fig. 18e). In addition, the survival of *S. typhimurium*- or *L. monocytogenes*-infected mice was significantly reduced following i.p. administration of PTZ (Fig. 7j and Supplementary Fig. 18f), suggesting that GABAergic inhibition significantly compromises the host defense against intracellular bacterial infections.

## Discussion

Our data demonstrate that GABAergic activation promotes antimicrobial host defenses, whereas GABAergic inhibition (under conditions that did not cause epilepsy or kindling, the progressive enhancement of seizure susceptibility) increased the overall susceptibility to infection and bacterial loads both in vitro and in vivo. Compared with extensive studies in brain and neural tissues, few studies have reported on the function of the GABAergic system in peripheral immune cells, such as macrophages and lymphocytes[7,8,18,19]. Herein, we focused on macrophages, the principal immune cells to combat intracellular bacterial infection, in GABAergic host defense. Importantly, macrophage depletion significantly affected the GABAergic defense system during infections.

We showed that macrophages and lung tissues synthesize GABA, whose levels were substantially decreased in the infected cells and serum following infection. GABA$_A$Rs are

heteropentameric ligand-gated chloride channels consisting of several classes of subunits that result in a number of possible subunit combinations in the cell membrane[35,36]. In the central nervous system, the assembly of GABA$_A$R heteropentamers consists of a combination of different subunit proteins encoded by 19 genes for GABA$_A$R subunits[37]. It is hypothesized that >800 GABA$_A$R subtypes exist, and that the α1β2γ2 subunit GABA$_A$R is the most abundant type in the brain[37]. Peritoneal macrophages contain functional GABA$_A$Rs, expressing mRNAs of the α1, α2, β3, and δ subunits[38]. Murine macrophages purified from spleens and lymph nodes exhibited GABA-evoked currents representing the functional GABA receptors, and also expressed β1 and ε subunits, whose levels differed in resting and lipopolysaccharide-stimulated cells[7]. Our data extend these previous findings by confirming the ability of 11 GABA$_A$R subunit transcripts and functional receptors containing the δ subunits to generate tonic currents in resting BMDMs. Given that the δ subunit confers high GABA sensitivity to GABA$_A$Rs in the brain[20], our results showing that GABA in the order of a few hundred μM activated tonic current suggest that a specific type of GABA$_A$Rs are functionally active in BMDMs. In human primary MDMs, we detected only three GABA$_A$R (GABRA1, GABRB1, and GABRR2), which expression profiles were different from those found in murine BMDMs, suggesting the differential effects of GABA on human and murine peripheral macrophages. Future studies are warranted to characterize the exact assembly and function of GABA$_A$Rs in peripheral macrophages from different species. In addition, our data first demonstrate that the levels of endogenous GABA and GABA$_A$R subunit transcripts were decreased by infection signals. Interestingly, the protein expression of GAD65/67 in macrophages increased following infection. There is speculation that the activities of GABA-synthesizing enzymes in macrophages can be increased by feedback activation during infection. Therefore, we examined the effects of exogenous GABA in the activation of host defense against intracellular bacterial infection.

Our results on the GABAergic activation of antimicrobial responses in macrophages and in vivo reveal the vital role of GABA$_A$R in innate immune responses against intracellular bacterial infections. As there are numerous GABA$_A$R subunits in macrophages, we were not able to determine the function of each subunit in antimicrobial responses. We thus examined the effects of functional GABA$_A$R in innate host defense by GABAergic activation with agonists or inhibition with antagonists. Notably, the administration of GABA or GABAergic agents inhibited, whereas GABAergic inhibition by the antagonists PTX and BIC enhanced, the number of granulomas and neutrophil infiltration in the lung tissues of mice infected with mycobacteria. Although

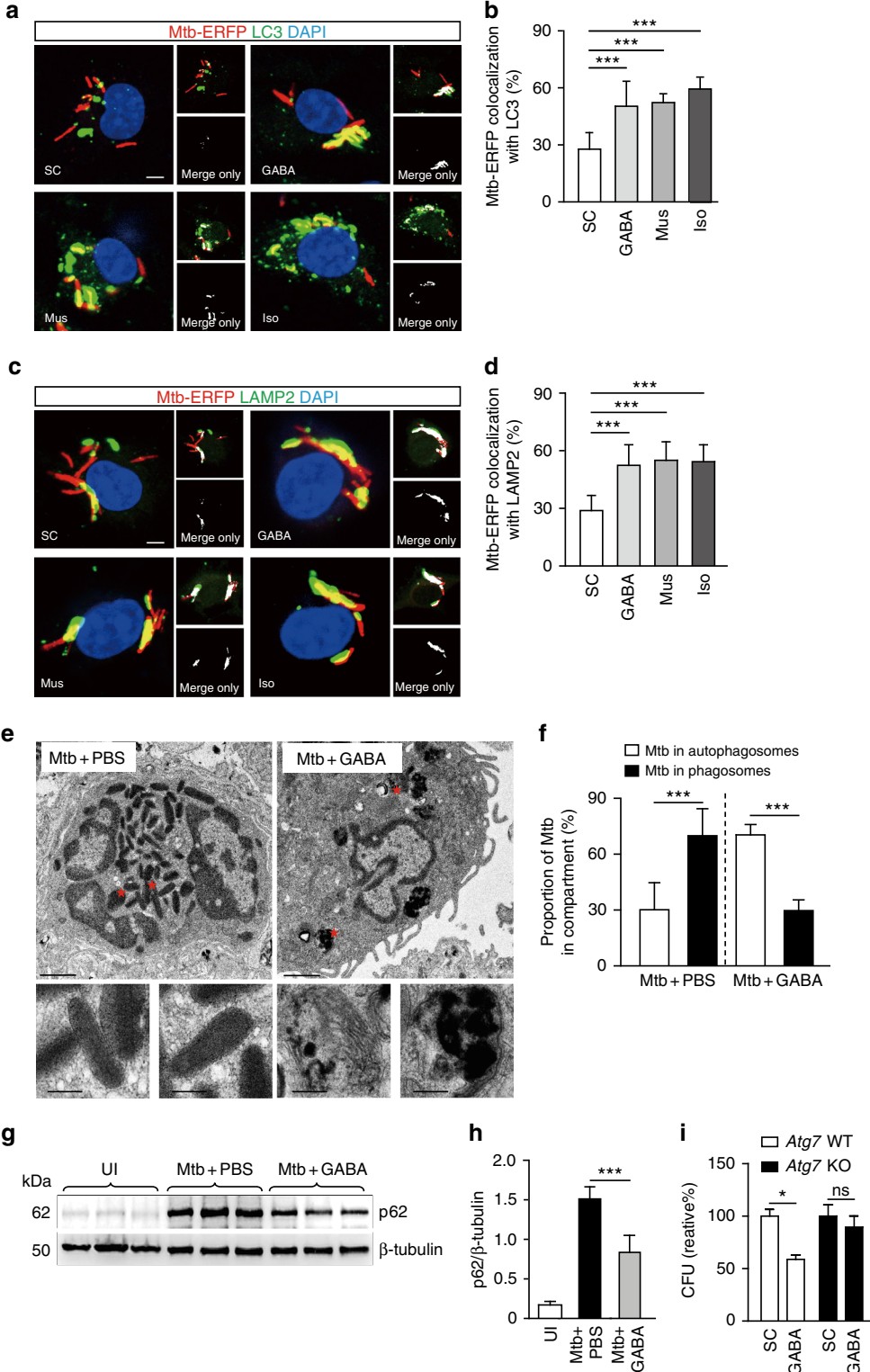

there are some experimental limitations in using an i.v. infection model, which may not represent aerosol infection well, we found that GABA treatment suppressed in vivo mycobacterial loads in intranasal (i.n.) and i.v. infection models. Pathologic inflammation and neutrophil infiltration, as well as excessive TNF production, are deleterious in mice and zebrafish models with tuberculosis (TB)[39–42]. We found that GABA treatment ameliorated inflammatory cytokine production and neutrophil infiltration in the infected sites (lungs) during Mtb infection. Because

neutrophils also express GABA$_A$Rs[43], GABA-mediated inhibition of neutrophil infiltration may be due to direct effects of GABA on neutrophils at the site of infection. Future studies are needed to clarify whether neutrophils directly respond to GABA and contribute to host defense during infection.

Additionally, earlier studies indicated the inhibitory effects of GABA on T-cell proliferation, proinflammatory Th1 responses, and delayed-type hypersensitivity (DTH) responses in vivo[44,45]. Moreover, treatment of honokiol, a phenolic GABA$_A$R agonist,

**Fig. 4** GABAergic autophagy activation promotes phagosomal maturation against mycobacterial infection. **a–d** BMDMs were infected with Mtb-ERFP (MOI of 5) for 4 h, incubated with GABA (100 μM), muscimol (Mus; 100 μM), or isoguvacine hydrochloride (Iso; 100 μM) for 18 h, then stained with LC3 (green; **a, b**) or LAMP2 (green; **c, d**), and DAPI (for nuclei; blue). **a** Cells were visualized by confocal microscopy. Scale bars, 5 μm. **b** Quantitative data of colocalization of Mtb-ERFP and LC3 per cell. **c** Cells were visualized by confocal microscopy. Scale bars, 5 μm. **d** Quantitative data of colocalization of Mtb-ERFP and LAMP2 per cell. **e–h** Mice were injected i.v. with Mtb (1 × 10$^6$ CFU), treated with PBS or GABA (daily i.p. 200 mg/kg), and monitored at 14 days post-infection (dpi). **e** Representative low- (top) and high-magnification (bottom) transmission electron micrographs of lung tissues. Scale bars, 1 μm (top) and 200 nm (bottom). **f** Quantitative data of phagosomes and autophagosomes containing Mtb. **g** Representative immunoblots for the expression of the indicated proteins from lung tissues. **h** The densitometric values for p62 were normalized to β-tubulin. **i** Atg7$^{f/f}$LysM-Cre$^-$ (Atg7 WT) and Atg7$^{f/f}$LysM-Cre$^+$ (Atg7 KO) BMDMs were infected with Mtb (MOI of 5) and then treated with GABA (100 μM). After 3 days, intracellular survival of Mtb was assessed by CFU assay. *$p < 0.05$, and ***$p < 0.001$. ns: not significant, UI: uninfected, SC: solvent control (0.1% PBS). Statistical significance was determined by one-way ANOVA (**b, d, h**), Mann–Whitney U-test (**f**), or two-way ANOVA (**i**). Data shown (means ± SEM) represent the combined results of triplicates from three independent experiments (**b, d, f, h, i**). Images are representative of three (**e, g**) or four (**a, c**) independent experiments

led to a marked inhibitory effect on Th1- and Th17-type inflammatory cytokines during lung inflammation[46]. Although we did not examine the effects of GABA on lymphocyte activation in mycobacterial or salmonella infection, we speculate that GABA treatment decreases inflammatory Th1 cells and regulates proinflammatory T-cell responses. Excessive stimulation of Th1 immunity in TB can result in pathologic inflammation[47]. In addition, type I interferons (T1-IFNs) and Th17 immunity have been recognized to be involved in TB pathogenesis[48,49]. Combined with the current data, GABAergic activation may contribute to host defenses and simultaneously control excessive inflammation associated with pathologic responses during bacterial infection. Future studies should be performed to clarify the exact function of GABAergic activation upon adaptive immune responses during intracellular bacterial infection.

Autophagy is a major catabolic process, removing protein aggregates and damaged organelles via lysosomal degradation[11], however, the function of GABA in autophagy regulation has remained largely unknown. Previous studies reported that the increase in GABA levels inhibited selective autophagy, that is, pexophagy and mitophagy, in yeast, but did not affect general autophagy activation[50]. Importantly, our data show that GABA or GABAergic activation induces autophagy, which in turn promotes phagosomal maturation and host antimicrobial responses to intracellular bacterial infections including Mtb and BCG. Our data are important to show that GABAergic activation triggered intracellular calcium release in macrophages, and that this was required for the induction of autophagy and phagosomal maturation against mycobacterial infection. In brain cells, GABAergic activation can lead to intracellular calcium influx via voltage-dependent calcium channels and/or N-methyl-D-aspartate receptors[51]. However, it remains largely unknown whether GABAergic stimulation activates intracellular calcium influx in peripheral immune cells. We found that GABA treatment of BMDMs increased the intracellular calcium release, which was completely blocked by BAPTA-AM, the chelator of intracellular calcium (Supplementary Fig. 14). Although we did not characterize how GABAergic activation triggers intracellular calcium release in macrophages, future studies are needed to investigate the molecular mechanisms by which intracellular calcium influx is increased by GABA treatment in macrophages. We further showed that AMPK was required for GABA-induced autophagy in macrophages. AMPK, a crucial sensor of energy metabolism, can be activated by an increase of AMP level and by Ca$^{2+}$/CaM-dependent kinase kinase 2[52]. As the activated AMPK phosphorylates ULK1 to lead to the activation of autophagy[53], AMPK plays key functions in the regulation of autophagy[54]. Accumulating evidence suggests that AMPK functions in the regulation of bacterial and viral infections, and that it is required for the enhancement of antibacterial autophagy against mycobacterial infection[27,28,55]. Combined with the current data, findings show that the Ca$^{2+}$-AMPK signaling pathway is critically involved in the

activation of GABAergic autophagy in macrophages during Mtb infection.

Further evidence has demonstrated that the Ca$^{2+}$-AMPK signaling pathway is essential for the transcriptional activation of several autophagy genes, including Gabarapl1, in BMDMs in response to GABA or GABAergic activation. It was noted that the GABARAPL1 protein, a member of the ATG8 family, is critically involved in phagosomal maturation and antimicrobial responses against intracellular Mtb. Earlier studies also showed that GABARAPL1 is involved in the formation of the autophagosome through association with autophagic vesicles by priming/delipidation by Atg4B[56,57]. It was further demonstrated that GABARAPL1 is associated with cellular metabolic functions and autophagic flux[29–31]. Moreover, recently, GABARAPL2 (Gate-16) was shown to be critically involved in the clearance of Toxoplasma through association with the small GTPase ADP-ribosylation factor 1[32]. However, it has not been investigated whether GABAergic signaling is linked to this GABARAP protein family member to regulate innate immune responses. Using Gabarapl1-null macrophages, our data demonstrate a key function of GABARAPL1 in the GABAergic activation of antibacterial autophagy and innate host defenses. Together, these data suggest that GABA activates GABAergic signaling via GABA$_A$R to trigger intracellular autophagic pathways through the Ca$^{2+}$-GABARAPL1 pathway to enhance host protection (summarized in Supplementary Fig. 17e).

Finally, we show that GABAergic inhibition increased mycobacterial, salmonella, and listerial loads in mice and mycobacterial loads in zebrafish. These data indicate that GABAergic signaling plays a conserved function in metazoan host defenses against intracellular bacterial infection. Because the experiments using GABA$_A$R antagonists were performed in the absence of GABA treatment per se, the basal levels of GABA and functional GABA$_A$R signaling are critical for appropriate induction of host defenses against intracellular bacterial infections. Additionally, GABA injection (200 mg/kg) caused no significant behavioral changes, as determined by the open field test, the rotarod test, and seizure sensitivity to electroshock (Supplementary Fig. 19), indicating that the maximal doses of GABA used in our study did not have any undesirable effects on the neurological system or GABA$_A$R signaling. In summary, we highlight a previously unappreciated role for the GABAergic system in the modulation of innate host defenses against intracellular bacterial infection. Such antibacterial effects indicate that GABA signaling may represent an intrinsic defense system protecting against a variety of infectious diseases. Recently, increasing efforts have been made to develop several autophagy-targeting agents as potential host-directed therapeutics to overcome drug resistance issues in a variety of infectious diseases, including

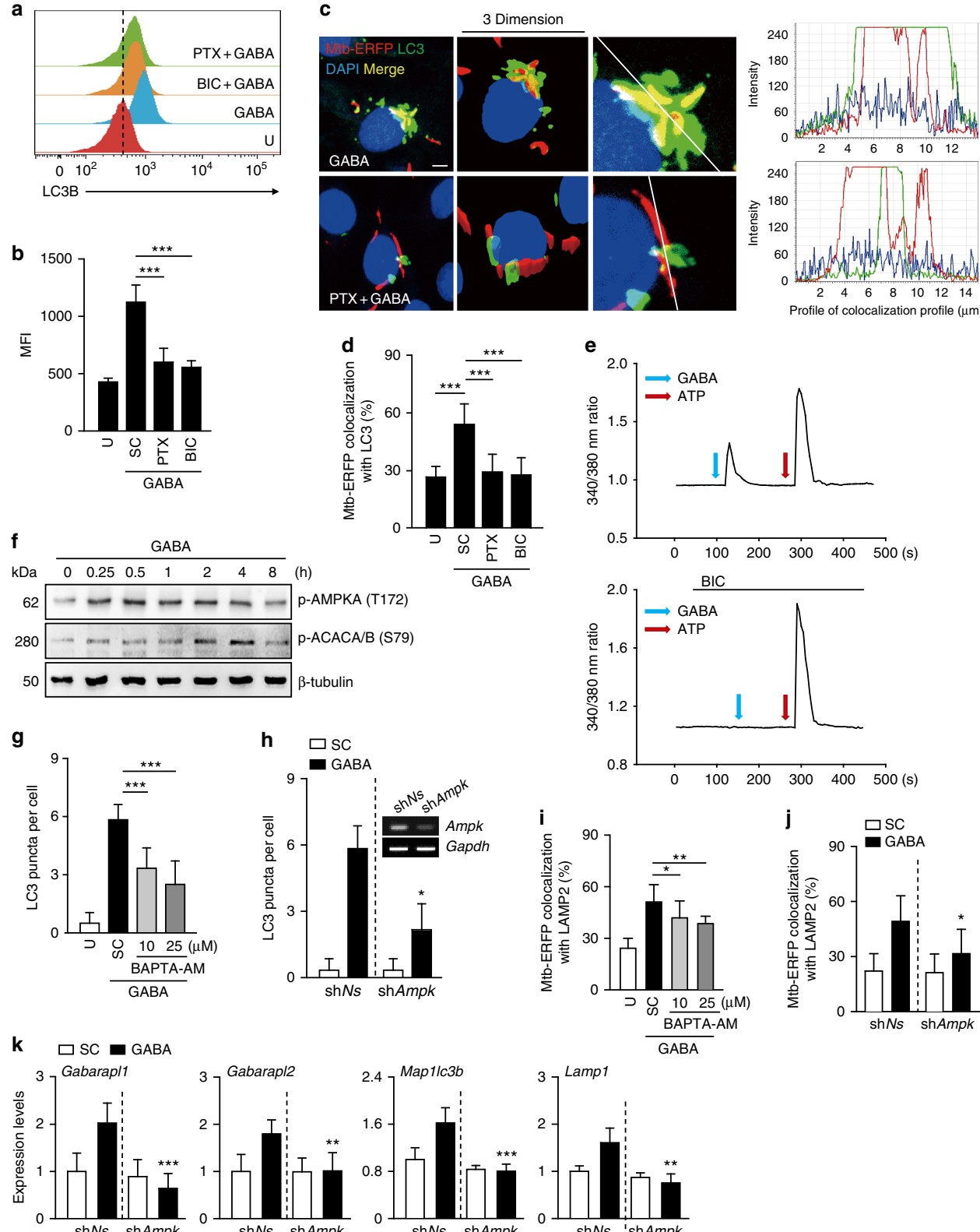

TB[58–60]. Our conceptual framework may be applied to identify additional drug-affected targets when seeking to treat intracellular bacterial infectious diseases.

## Methods

**Mice and zebrafish**. C57BL/6 mice (sex-matched) aged 6–8 weeks with a wild-type (WT) background were purchased from Samtako Bio Korea (Gyeonggi-do, Korea) to generate transgenic myeloid lineage cell-specific *Atg7*-deficient mice (*Atg7*fl/fl LysM-Cre+) using the Cre/loxP recombination system[61]. Mice were maintained under specific pathogen-free conditions. *Gabarapl1* KO mice were generated by CRISPR genome editing as described previously[32]. The zebrafish (AB line) and their embryos were handled and staged in accordance with standard protocols[62]. For zebrafish anesthesia procedures, embryos were immersed in a 270 mg/L Tricaine solution in osmotic water. Animal numbers were predetermined based on pilot studies and sample sizes were similar to generally employed in the field. No data were excluded from the analysis.

**Fig. 5** GABAergic autophagy is mediated via GABA$_A$R, intracellular calcium release, and AMPK signaling. **a, b** BMDMs were pretreated with bicuculline (BIC; 100 µM) or picrotoxin (PTX; 100 µM) for 1 h, and treated with GABA (100 µM). **a** Flow cytometric analysis for LC3B expression (horizontal axis, FITC intensity). Representative gating strategy is shown in Supplementary Fig. 5. **b** Average MIFs of LC3B. **c, d** BMDMs were infected with Mtb-ERFP in the same condition with **a, b**, and then stained with LC3 (green) and DAPI (nuclei; blue). **c** Confocal z-stack images. Scale bars, 5 µm (left). Tracing of colocalization (right). **d** Quantitative data of colocalization of Mtb-ERFP and LC3. **e** BMDMs were stained with Fura-2 AM and incubated with GABA (100 µM) in the absence (top) or presence (bottom) of BIC (100 µM). **f** BMDMs were incubated with GABA (100 µM) for the indicated times. Representative immunoblots for the indicated protein expression. **g, h** Quantitative data of LC3 punctate analysis. **g** BMDMs were treated with BAPTA-AM (10 or 25 µM) before GABA (100 µM) incubation. **h, j, k** BMDMs were transduced with lentivirus expressing sh*Ns* or sh*Ampk*, and then treated with GABA (100 µM), followed by LC3 staining (**h**; semiquantitative PCR analysis for transduction efficiency, inset); Quantitative analysis of colocalization of Mtb-ERFP and LAMP2 (**j**); qRT-PCR analysis for autophagy genes (**k**). **i** Quantitative analysis of colocalization of Mtb-ERFP and LAMP2. BMDMs infected with Mtb-ERFP were incubated with GABA (100 µM) in the presence or absence of BAPTA-AM (10 or 25 µM). *$p < 0.05$, **$p < 0.01$, and ***$p < 0.001$. U: untreated, SC: solvent control (0.05% EtOH, for **b, d**; 0.1% PBS, for **h, j, k**; 0.1% dimethyl sulfoxide, for **g, i**). Statistical significance was determined by one-way ANOVA (**b, d, g, i**), and two-way ANOVA (**h, j, k**). Data shown (means ± SEM) represent the combined results of duplicates from two (**b**) or three (**d, g, h–k**) experiments. Images are representative of two (**a**) or three (**c, e, f**) independent experiments

**Cells**. Primary BMDMs isolated from C57BL/6 mice and cells of the murine macrophage cell line RAW264.7 (ATCC, TIB-71) were cultured in Dulbecco's modified Eagle's medium (DMEM; Lonza) containing 10% fetal bovine serum (FBS; Lonza), penicillin (100 IU/ml), and streptomycin (100 µg/ml). BMDMs were differentiated for 3–5 days in the presence of macrophage colony-stimulating factor (M-CSF; R&D Systems). Phoenix AMPHO (ATCC, CRL-3213) cells were maintained in DMEM and incubated in a 37 °C humidified atmosphere with 5% CO$_2$. RAW264.7 and Phoenix AMPHO cells, which were authenticated by the provider. Cell lines were routinely tested for mycoplasma using a commercially available kit (MycoAlert, Lonza). For this study, we recruited six healthy Korean volunteers in Daejeon, Korea; six healthy volunteers (male/female 3/3; between age 30 and 52 (mean 41)). The healthy volunteers had no past and current diagnosis of specific diseases. Human peripheral blood mononuclear cells were separated from heparinized venous blood of healthy donors by gradient centrifugation on Ficoll-Hypaque (Lymphoprep; Axis-Shield, 1114545) at room temperature[63]. After cells were allowed to adhere to culture plate for 1 h at 37 °C, the non-adherent cells were removed by vigorous washing with phosphate-buffered saline (PBS). For macrophage differentiation, adherent monocytes were incubated in RPMI-1640 (Lonza, 12-702 F) with 10% pooled human serum (Lonza, 14-402), 1% L-glutamine, 50 IU/ml penicillin, and 50 µg/ml streptomycin for 1 h at 37 °C, and non-adherent cells were removed. Human MDMs were prepared by culturing peripheral blood monocytes for 4 days in the presence of 4 ng/ml human CSF/M-CSF (Sigma-Aldrich, M6518)[55].

**Bacterial strains and culture**. Mtb H37Rv was kindly provided by Dr. R. L. Friedman (University of Arizona, Tucson, AZ, USA). *M. bovis* BCG was obtained from the Korean Institute of Tuberculosis (Osong, Korea). Mtb and BCG were grown at 37 °C with shaking in Middlebrook 7H9 broth (Difco, 271310) supplemented with 0.5% glycerol, 0.05% Tween-80 (Sigma-Aldrich), and oleic albumin dextrose catalase (OADC; BD Biosciences, 212240). For Mtb- or BCG-expressing enhanced red fluorescent protein (ERFP) strains, Mtb- or BCG-ERFP strains were grown in Middlebrook 7H9 medium supplemented with OADC and 50 µg/ml kanamycin (Sigma-Aldrich, 60615)[64]. All mycobacterial suspensions were aliquoted and stored at −80 °C. For all experiments, mid-log-phase bacteria (absorbance 0.4) were used. Representative vials were thawed and colony-forming units (CFUs) enumerated by serially diluting and plating on Middlebrook 7H10 agar (Difco, 262710).

The *S. typhimurium* strain SL1344 was kindly provided by H. E. Choy (Chonnam National University, Korea) and grown in Luria-Bertani (LB; Lab-Pharm-Service solutions) broth at 37 °C for 12 h. Bacterial cultures reaching saturation density were diluted (1:100) and cultured for 4 h to reach the mid-log growth phase (absorbance 0.4). Bacterial cultures were centrifuged and washed with PBS (Sigma-Aldrich) twice before use. Serial dilutions were plated on an LB plate to confirm the number of bacteria.

WT *M. marinum* (strain TMC 1218—ATCC # 927) constitutively expressing tdTomato (red fluorescence) was cultured in 7H9 broth with OADC and 0.2% Tween-80, at 30 °C in the dark without agitation. *M. marinum* shuttle expression plasmid pTEC27 was from Addgene (deposited by Lalita Ramakrishnan). *M. marinum* was transformed with the recombinant plasmid pMSP12::tdTomato by electroporation; then, hygromycin-resistant clones were selected in 7H9 broth supplemented with hygromycin for further experiments.

**CFU assay**. To assess intracellular bacterial viability, cells were infected with Mtb for 4 h. The infected cells were washed three times with PBS to remove extracellular bacteria. The infected cells were incubated for the indicated periods and then lysed with 0.3% saponin (Sigma-Aldrich) in distilled water to release intracellular bacteria. Thereafter, bacteria were harvested and inoculated onto Middlebrook 7H10 agar with OADC. Plates were incubated for 3 weeks, and colonies were counted.

**Mouse infection**. For mycobacterial infection, frozen Mtb or BCG were thawed and centrifuged. The pellet was resuspended in PBS plus 0.05% Tween-80 (PBST) before infection. Mice were injected with Mtb (i.v., $1 \times 10^5$ or $10^6$ CFU/mouse) or BCG (i.v., $1 \times 10^7$ CFU/mouse; or i.n., $2 \times 10^6$ CFU/mouse), or BCG-ERFP (i.n., $5 \times 10^5$ CFU/mouse). For measurement of the bacterial burden in the lung, spleen, and liver, mice were sacrificed at the indicated times after Mtb or BCG infection, tissues were homogenized in PBST, and serial dilutions of the homogenates were plated on 7H10 agar plates, with colonies counted 2–3 weeks later. Mice were orally infected with $2 \times 10^7$ or $1 \times 10^8$ CFU of *S. typhimurium* and then killed after the indicated time. For measurement of the bacterial burdens in liver, spleen, and MLN, these were collected and the tissues were homogenized in sterile PBST, and serial dilutions of the homogenates were plated on LB agar plates, with colonies counted 24 h later. GABA (200 mg per kg body weight) or PTZ (20 mg or 40 mg per kg body weight) was used throughout the experiments. Before the Mtb, BCG, or *S. typhimurium* infection, PBS or GABA was i.p. injected into mice once daily for 7 days. After the infection, PBS or GABA were injected once daily for the indicated time. PBS or PTZ were i.p. injected into mice 6 h or 7 days (once every two days) before Mtb, BCG, or *S. typhimurium* infection.

**Systemic infections in zebrafish embryos**. *M. marinum* were prepared and microinjected in zebrafish embryos, in accordance with previously reported procedures[65]. Briefly, systemic infections were carried by the injection of ~200 CFU into the caudal vein of 30 h post-fertilization embryos.

**Antibodies and reagents**. Anti-GABA (1:400 diluted; A2052), anti-LC3A/B (1:1000 diluted for immunoblotting; L8918), and anti-p62 (1:1000 diluted; P0067) were purchased from Sigma-Aldrich. Anti-LAMP2 (1:400 diluted; sc-5571, sc-18822) was purchased from Santa Cruz Biotechnology. Anti-GABARAPL1 (1:400 diluted; ab86497) and anti-β-tubulin (1:1000 diluted; ab6046) were purchased from Abcam. Anti-LC3A/B (1:400 diluted for immunofluorescence; PM036) was purchased from MBL International. Anti-LC3B (1:100 diluted for flow cytometry; 2775s), anti-phospho-AMPKA (1:1000 diluted; 2535s), anti-phospho-ACACA/B (1:1000 diluted; 3661s) were purchased from Cell Signaling. Alexa Fluor 488-conjugated anti-rabbit IgG (1:400 diluted; A17041), Alexa Fluor 594-conjugated anti-rabbit IgG (1:400 diluted; A21207), and Alexa Fluor 568-conjugated anti-mouse IgG (1:400 diluted; A11004) were purchased from Invitrogen. Anti-GAD65 (1:400 diluted; PA5-22260) was purchased from Thermo Fisher Scientific. Muscimol (M1523), isoguvacine hydrochloride (G002), GABA (A2129), BIC (B7686), 4′-6-diamidino-2-phenylindole dihydrochloride (DAPI; D9542), PTX (M7514), ATP (A26209), and PTZ (P6500) were purchased from Sigma-Aldrich. BAPTA-AM (196419) was purchased from Calbiochem.

**Immunoblot analysis**. Cells or tissues were lysed in radioimmunoprecipitation assay (RIPA) buffer (50 mM Tris-HCl, pH 7.5, 2 mM EDTA, 150 mM NaCl, 0.1% sodium dodecyl sulfate (SDS), 1% sodium deoxycholate, and 1% Triton X-100) supplemented with protease inhibitor cocktail (Roche). Protein extracts were boiled in 1× SDS sample buffer and subjected to immunoblotting analysis. Proteins were separated by 12 or 15% SDS-polyacrylamide gel electrophoresis (PAGE) and transferred to a polyvinylidene difluoride membrane (Millipore). Membranes were blocked in 5% nonfat milk in PBST (3.2 mM Na$_2$HPO$_4$, 0.5 mM KH$_2$PO$_4$, 1.3 mM KCl, 135 mM NaCl, 0.05% Tween 20 [Sigma-Aldrich], pH 7.4) for 1 h and incubated with the primary antibodies. After incubation with the appropriate antibodies, immunoreactive band analysis was performed using an ECL reagent (Millipore), and bands were detected using a UVitec Alliance mini-chemiluminescence device (UVitec, UK). Uncropped original scan images are shown in Supplementary Fig. 20.

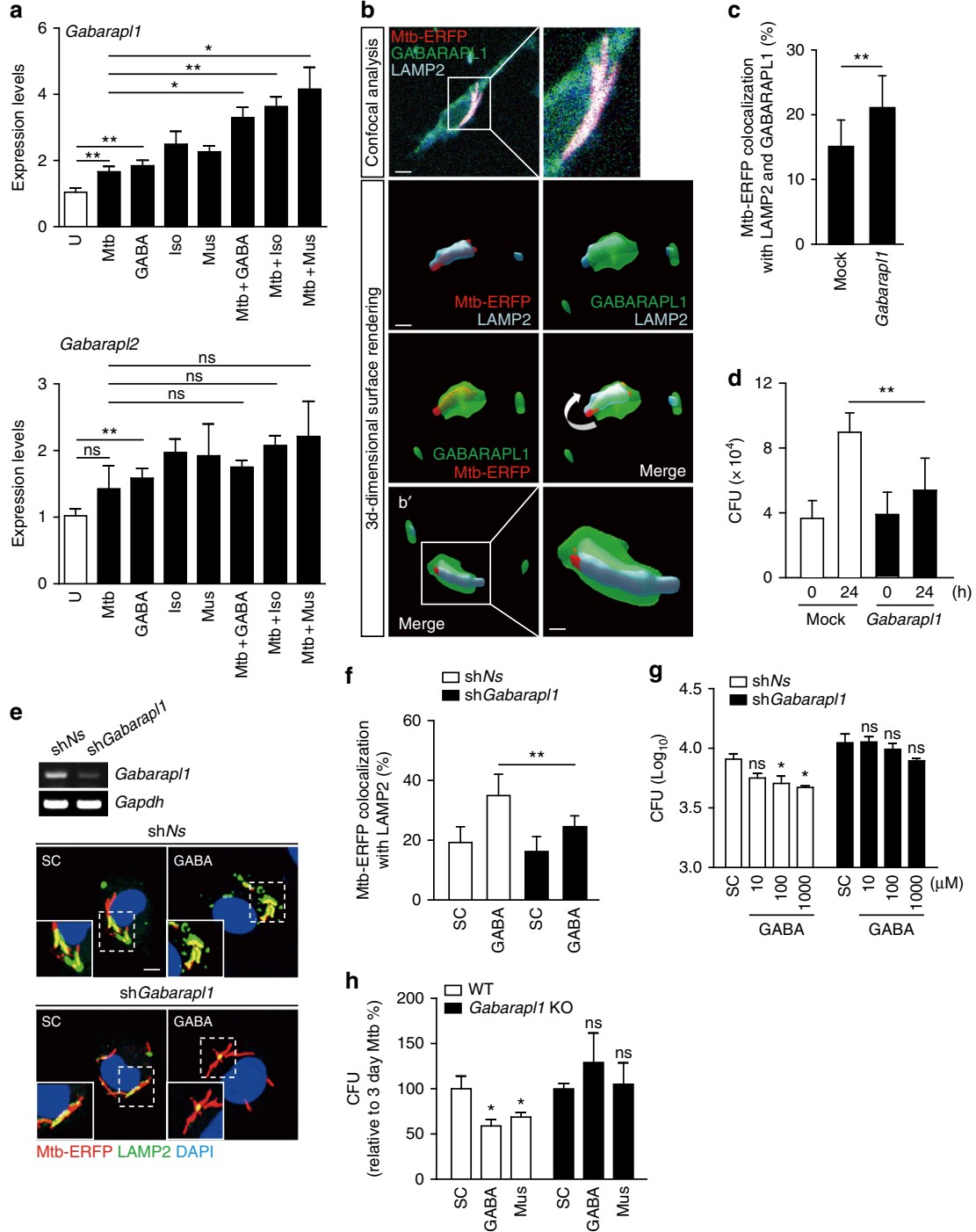

**Fig. 6** GABARAPL1 is essential for phagosomal maturation and antimicrobial responses in macrophages. **a** BMDMs infected with Mtb (4 h) were incubated with GABA (100 μM), muscimol (Mus; 100 μM), or isoguvacine hydrochloride (Iso; 100 μM) for 6 h, followed by qRT-PCR analysis for *Gabarapl1* and *Gabarapl2*. **b**, **c** RAW264.7 cells were transfected with mock or EGFP-*Gabarapl1* plasmid for 24 h and infected with Mtb-ERFP. Cells were fixed and stained with LAMP2 (cyan). **b** Cells were visualized by confocal microscopy. Confocal z-stack images were obtained and reconstituted to three dimensions. Scale bars: 10 μm (top), 5 μm (middle), and 2.5 μm (bottom). **b'** shows different angles of the models. **c** Quantitative data of colocalization of Mtb-ERFP, LAMP2, and GABARAPL1 per cell. **d** RAW264.7 cells were transfected with mock or *Gabarapl1* plasmid for 24 h and infected with Mtb for the indicated times. Intracellular survival of Mtb assessed by CFU assay. **e**–**g** BMDMs were transduced with lentivirus expressing sh*Ns* or sh*Gabarapl1* using polybrene (8 μg/ml). **e**, **f** After 36 h, BMDMs were infected with Mtb-ERFP for 4 h and incubated with GABA (100 μM) for 18 h. **e** Semiquantitative PCR analysis was performed to assess transduction efficiency (top). Cells were visualized by confocal microscopy (bottom). Scale bars, 5 μm. **f** Quantitative data of colocalization of Mtb-ERFP and LAMP2 per cell. **g** Intracellular survival of Mtb assessed in BMDMs treated with GABA (10, 100, and 1000 μM) for 3 days. **h** BMDMs from WT or *Gabarapl1* KO were infected with Mtb and treated with GABA (100 μM) or Mus (100 μM) for 3 days. Intracellular survival of Mtb assessed by CFU assay. *$p < 0.05$ and **$p < 0.01$. U: untreated, SC: solvent control (0.1% PBS), ns: not significant. Statistical significance was determined by one-way ANOVA (**a**, **g**, **h**), Mann–Whitney *U*-test (**c**), and two-way ANOVA (**d**, **f**). Data shown (means ± SEM) represent the combined results of duplicates (**a**, **h**) or triplicates (**c**, **d**, **f**, **g**) from three independent experiments. Images are representative of three independent experiments (**b**, **e**)

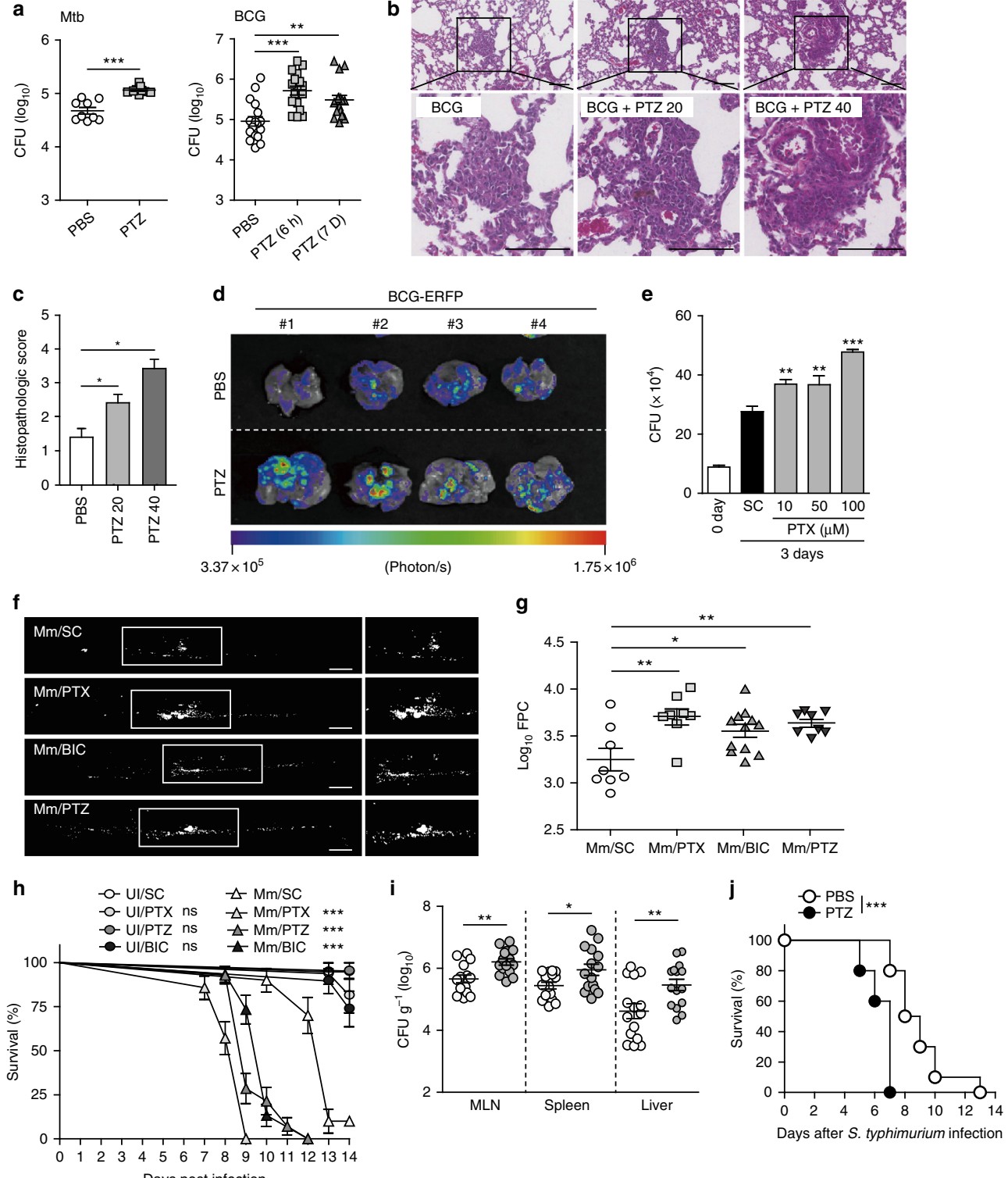

**Voltage-clamp recordings and analysis**. Patch-clamp recordings were performed on the primary BMDMs. Whole-cell configuration was established and currents were recorded at a holding potential of −70 mV at 32 °C using an Axopatch 200B amplifier (Molecular Devices, USA). Current output filtered at 2 kHz with the internal filter was digitized at 10 kHz using a Digidata 1322A digitizer (Molecular Devices). Patch pipettes were filled with a high-Cl− solution with the following composition (in mM): 140 KCl, 10 HEPES, 5 Mg$^{2+}$ATP, 0.9 MgCl$_2$, and 10 EGTA. Y-tube method[66] was used for rapid and focal GABA application onto macrophages in the extracellular recording solution (in mM): 126 NaCl, 2.5 KCl, 1 MgCl$_2$, 1.8 CaCl$_2$, 26 NaHCO$_3$, 1.25 NaH$_2$PO$_4$, pH 7.4, saturated with 95% O$_2$−5% CO$_2$ and 297 mOsm. Whole-cell currents were acquired and analyzed with pClamp

9.0 (Molecular Devices). GABA-induced current was defined as the difference between the holding current before and after the application of GABA.

**Lentiviral shRNA production and transduction**. For the silencing of mouse *Gabarapl1* or *Ampk* in primary cells, packaging plasmids (pRSV-Rev; 12253, pMD2. G; 12259 and pMDLg/pRRE; 12251, purchased from Addgene, deposited by Dr. Didier Trono) and pLKO.1-based target shRNA plasmids (Santa Cruz Biotechnology, *Gabarapl1*; sc-145303-SH or *Ampka*; sc-29674-SH) were co-transfected into 293T cells using Lipofectamine 2000 (Invitrogen, 11668019). After 72 h, virus-containing medium was collected and filtered. BMDMs in DMEM

**Fig. 7** GABAergic inhibition decreases host protection to intracellular bacterial infection. **a**, **b** Mice were injected with PBS or pentylenetetrazol (PTZ; 40 mg/kg; i.p.) 6 h or 7 days before Mtb (i.v., $1 \times 10^5$ CFU; $n = 9$ per group) or BCG infection (i.n., $2 \times 10^6$ CFU; $n = 8$ per group). **a** CFU assay in the lungs (Mtb, 14 dpi; BCG, 7 dpi). **b** Representative H&E-stained images from lung tissues at 7 dpi. Mice were injected with PBS or PTZ (i.p., 20 or 40 mg/kg) before BCG infection. Scale bars, 100 μm. **c** Quantitative analysis of histopathology. **d** Representative in vivo imaging of BCG-ERFP-infected lungs (i.n., $5 \times 10^5$ CFU) from mice ($n = 4$ per group) treated with PBS or PTZ (40 mg/kg, 6 h; 7 dpi). **e** Intracellular survival assay for Mtb in BMDMs treated with picrotoxin (PTX; 10, 50, or 100 μM) for 3 days. **f–h** Zebrafish embryos were i.v. infected with *M. marinum*-tdTomato (Mm; 200 CFU) and treated with BIC (50 μM), PTX (5 μM), or PTZ (50 μM), at 3 dpi. **f** Representative fluorescence images. Scale bars, 100 μm. **g** Bacterial loads (FPC; fluorescent pixel counts) of **f**. **h** Survival of *M. marinum*-infected or PBS-injected embryos (treated with BIC, PTX, or PTZ; $n = 22$–32). **i** and **j** PBS or PTZ (40 mg/kg) were injected i.p. into mice ($n = 10$ per group) 6 h before oral infection with *S. typhimurium* ($2 \times 10^7$ CFU). **i** Viable cell count of *S. typhimurium* in mice at 5 dpi. **j** Survival of mice. $*p < 0.05$, $**p < 0.01$, and $***p < 0.001$. UI: uninfected, SC: solvent control (0.05% EtOH, for d), ns: not significant. Statistical significance was determined by unpaired *t*-test (**a** Mtb, **i**), one-way ANOVA (**a** BCG, **c**, **e**, **g**), and log-rank (Mantel–Cox) test (**h**, **j**). Data shown (means ± SEM) represent the combined results of triplicates from three independent experiments (**e**). Values are pooled from three (**a**, **c**, **g–j**) independent experiments. Images are representative of three independent experiments (**b**, **d**, **f**). Each symbol represents one animal (**a**, **g**, **i**)

---

medium supplemented with 10% FBS were seeded into 24-well plates and then infected with lentiviral vectors in the presence of 8 μg/ml polybrene (Sigma-Aldrich). On the following day, the medium was replaced with fresh medium. After 3 days, transduction efficiency was analyzed by PCR.

**Plasmid construction**. The DNA fragment corresponding to the coding sequences of the m*Gabarapl1* gene was amplified by PCR and subcloned into pcDNA3-Flag vector (Addgene, 20011; deposited by Stephen Smale) between the *Eco*RI and *Bam*HI sites and pEGFP-vector (Clontech) between the *Eco*RI and *Bam*HI sites. All plasmid constructs were sequenced using the ABI PRISM 377 automatic DNA sequencer (Applied Biosystems, Inc., Carlsbad, CA, USA) to verify 100% correspondence with the original sequence.

**Generation of a tandem LC3B retroviral vector**. The production of a tandem LC3B retroviral vector (mCherry-EGFP-LC3B) for the measurement of autophagic flux was performed[67]. Briefly, Phoenix amphotropic cells were seeded into a six-well plate and co-transfected with 0.75 μg of packaging plasmid pCL-Eco (Addgene), 0.25 μg of envelope plasmid pMDG (Addgene), and 1 μg of pBABE-puro mCherry-EGFP-LC3B plasmid (Addgene, 22418) using Lipofectamine 2000. After 6 h, the medium was replaced with fresh culture medium. The retrovirus-containing medium was harvested at 24 h and 48 h post-transfection and filtered through a 0.45-μm syringe filter.

**RNAseq analysis**. BMDMs were treated with GABA for the indicated times and washed with PBS. Total RNA was extracted from each sample using TRIzol (Invitrogen). An RNA sequencing library was generated using Nextflex Rapid Directional qRNA-Seq Kit, in accordance with the user's instruction manual (Bioo Scientific, Austin, TX, USA). Briefly, mRNA was purified from total RNA using Oligo (dT) beads and was chemically fragmented. After double-strand complementary DNA (cDNA) synthesis of the fragmented mRNA, adenylation of the 3′-end, sequencing adapter ligation, Uracil-DNA Glycosylase (UDG) treatment, and PCR amplification were performed, followed by DNA purification with magnetic beads. Finally, the amplified library was checked with BioAnalyzer 2100 (Agilent, CA, USA) and then applied for sequencing template preparation. The HiSeq2500 platform was utilized to generate 100-bp paired-end sequencing reads (Illumina, CA, USA).

**Measurement of intracellular $Ca^{2+}$**. BMDMs grown on coverslips were loaded with the $Ca^{2+}$ indicator Fura-2 AM (10 μM; Invitrogen, 18064) in Hank's balanced-salt solution, in accordance with the manufacturer's instructions (Invitrogen). Cells were subsequently washed twice with Hank's balanced-salt solution and were treated with GABA. Fura-2 AM was excited with the 380-nm line (calcium free) and 340-nm line (calcium complex) of an argon laser, and emission was detected at 510 nm.

**Autophagy analysis**. To quantify autophagy, LC3 punctate dots were identified using ImageJ software. Each condition was assayed in triplicate, and at least 100 cells per well were counted. To quantify the percentages of mycobacterial phagosomes, autophagosomes (LC3), and lysosomes (LAMP2), Mtb-infected cells were visualized directly by confocal laser-scanning microscopy, and the images were captured using Leica software (LAS X; Leica). Quantification of mycobacterial colocalization with autophagosomes and lysosomes was performed by counting the red (noncolocalized) and yellow (colocalized) mycobacteria. A minimum of 300 bacterial phagosomes or autophagosome and lysosome markers were analyzed per coverslip for each experiment. Each experiment was completed on triplicate coverslips and the results are expressed as the mean and standard deviation. Images of dynamic cell colocalization were recorded as vertical z-stacks. LAS X small 2.0 and Adobe Photoshop 7 (Adobe Systems) were used for image processing.

**RNA extraction and RT-PCR analysis**. Total RNA from homogenized lung or cell was isolated using TRIzol reagent (Thermo Fisher Scientific, 15596-026), in accordance with the manufacturer's instructions. After RNA quantitation, cDNA was synthesized by reverse transcription using the reverse transcriptase premix (Elpis Biotech, EBT-1515). For semiquantitative RT-PCR, the PCR was performed using the Prime Taq Premix (GeNet Bio, G-3000), and samples were amplified for 30 cycles as follows: 95 °C for 30 s, 55 °C for 30 s, and 72 °C for 30 s. For real-time RT-PCR analysis, real-time PCR was carried out using cDNA, primers, and SYBR Green master mix (Qiagen, 204074). Reactions were run on a Rotor-Gene Q 2plex system (Qiagen, Germany). To analyze real-time PCR data, relative quantification using the $2^{\Delta\Delta}$ threshold cycle (Ct) method using *Gapdh* for normalization. Data are expressed as relative fold changes. The primer sequences used in this article were shown in Supplementary Table 1.

**Immunofluorescence and confocal microscopy**. Immunofluorescence and confocal microscopy were performed[68]. After the appropriate treatment, cells on coverslips were washed three times with PBS, fixed with 4% paraformaldehyde for 15 min, permeabilized with 0.25% Triton X-100 (Sigma-Aldrich) for 10 min, and incubated with primary antibodies for 2 h at room temperature. Cells were washed with PBS to remove excess primary antibodies and then incubated with secondary antibodies for 1 h at room temperature. Nuclei were stained with DAPI for 2 min. After mounting, fluorescence images were acquired using a confocal laser-scanning microscope (TCS SP8; Leica, Wetzlar, Germany), with constant excitation, emission, pinhole, and exposure time parameters. To quantify bacterial loads, quantification of fluorescent *M. marinum* infection was performed using images of individual embryos[65].

**Enzyme-linked immunosorbent assay (ELISA)**. For the measurement of GABA production in lung lysates of mice or supernatants of BMDMs infected with Mtb, ELISA was performed. Lung lysates or supernatants were analyzed using a Mouse ELISA Kit to detect GABA (Novatein Biosciences; NB-E20434). All assays were performed as recommended by the manufacturer.

**Flow cytometry**. BMDMs were analyzed by flow cytometry for LC3B or GAD65 using a FACS Canto II flow cytometer, as indicated by the manufacturer (Becton Dickinson, San Jose, CA, USA). After two washes with PBS, cells were fixed in 4% paraformaldehyde for 10 min at 37 °C or permeabilized with 0.25% Triton X-100 in PBS for 10 min. Cells were stained with primary antibodies for 1–2 h at 4 °C (1:100) and the secondary antibodies for 1 h on ice. After two washes with PBS, cells were fixed in 4% paraformaldehyde and assayed immediately. Flow cytometry data were collected and analyzed using FlowJo software (Tree Star, Ashland, OR, USA).

**Transmission electron microscopy**. For transmission electron microscopy analysis, infected lung tissues or BMDMs were washed with PBS, fixed with 4% paraformaldehyde and 2% glutaraldehyde in 0.1 M sodium cacodylate buffer (pH 7.4) for 1 h, post-fixed in 1% osmium tetroxide and 0.5% potassium ferricyanide in cacodylate buffer for 1 h, embedded in resin, and cured at 80 °C for 24 h. Ultrathin sections (70–80 nm) were cut using an ultramicrotome (RMC MT6000-XL), stained with uranyl acetate and lead citrate, and examined using a Tecnai G2 Spirit Twin transmission electron microscope (FEI Co., Hillsboro, OR, USA) and a JEM ARM 1300 S high-voltage electron microscope (JEOL, Tokyo, Japan).

**Histology**. For histopathology, lung samples were fixed in 10% formalin and embedded in paraffin wax. Paraffin sections (4 μm) were then cut and stained with hematoxylin and eosin (H&E). H&E-stained sections were scanned with an Aperio digital pathology slide scanner (Leica) and imaged using an Aperio ScanScope® CS System. The histopathological score was graded for severity by scanning multiple random fields in six sections per mouse. An overall histopathological score was

assigned to lung tissue in each animal based on the extent of granulomatous inflammation as follows: 0 = no lesion, 1 = minimal lesion (1–10% of tissue in section involved), 2 = mild lesion (11–30% involved), 3 = moderate lesion (30–50% involved), 4 = marked lesion (50–80% involved), and 5 = severe lesion (> 80% involved)[69]. For immunohistochemical (IHC) staining, lung paraffin sections (4 μm) were cut and immunostained with antibodies specific for GABA. IHC-stained lung tissue slides were examined using a confocal laser-scanning microscope. Confocal images were taken with a Leica TCS SP8 confocal system and were processed with the Leica LAS AF Lite program.

**Optical bioluminescence imaging**. Mice were infected i.n. with BCG-ERFP (for GABA effect, $2 \times 10^6$ CFU/mouse; for PTZ effect, $5 \times 10^5$ CFU/mouse), treated with PBS or GABA, and monitored at 7 dpi. To confirm bacterial colonization of lung, ex vivo fluorescence imaging was performed using an optical imaging system (Berthold Technologies, Night OWL LB983). BCG-ERFP bacteria were detected with an excitation/emission wavelength of 530/600 nm with exposure of 60 s. Data were analyzed with IndiGo software (Berthold Technologies).

**Ethics statement**. All mice were bred and housed for experiments in accordance with the guidelines of Chungnam National University, School of Medicine, and Korea Research Institute of Bioscience and Biotechnology in biosafety level 3 laboratory facilities. All animal experimental methods and procedures were performed in accordance with the relevant ethical guidelines and regulations. This study was approved by the Institutional Research and Ethics Committee at Chungnam National University, School of Medicine (CNU-00944; Daejeon, Korea) and Korea Research Institute of Bioscience and Biotechnology (project license: KCDC-15-3-01). All animal procedures were conducted in accordance with the guidelines of the Korean Food and Drug Administration. The participants in the study were healthy volunteers recruited from Daejeon city based on flyers. The study was approved by the institutional review board of the Chungnam National University Hospital Bioethics Committee (project license: CNUH 2014-04-039-009) and informed consent was obtained from all donors.

**Statistical analysis**. All statistical tests were performed with GraphPad Prism version 5, and normality was assessed using D'Agostino and Pearson omnibus normality test. The two-tailed unpaired $t$-test (for parametric data) or Mann–Whitney $U$-test (for nonparametric data) were used for comparisons of two groups, one-way analysis of variance (ANOVA; Dunnett's test) was performed for comparisons of multiple groups. For comparison of the magnitude of changes in different conditions, two-way ANOVA with Bonferroni post-tests was used. Survival studies were analyzed with the Mantel–Cox log-rank test. Differences were considered significant (*$p < 0.05$, **$p < 0.01$, ***$p < 0.001$, ns: not significant). Data are presented as mean ± SEM. Animals were studied in group sizes as described in figure legends.

## Data availability

The array data support the findings of this study have been deposited in the NCBI Gene Expression Omnibus (GEO) database with the series accession number GSE113999. The other data that support the findings of this study are available from the corresponding authors upon request.

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

## Acknowledgements

We acknowledge Dr. R. L. Friedman for kindly providing mycobacterial strains. We would like to thank Dr. C. S. Yang for kindly providing *L. monocytogenes*. We thank Dr. T. S. Kim, Dr. H.W. Suh, Dr. S. Y. Kim, and J. H. Choe for excellent technical assistance. We also thank Dr. C. J. Lee and Dr. P. Silwal, for critical discussion and reading of the paper. Finally, we thank all of Dr. E. K. Jo's lab members for their helpful discussions. This work was supported by the National Research Foundation of Korea (NRF) Grant funded by the Korean Government (MSIP) (no. 2017R1A5A2015385 to E.-K.J. and no. 2015R1D1A1A02059430 to J.B.P). This work was supported by the National Research Foundation of Korea (NRF) grant funded by the Korea government (MSIP) (n. NRF-2015M3C9A2054326). This research was supported by a grant of the Korea Health Technology R&D Project through the Korea Health Industry Development Institute (KHIDI), funded by the Ministry of Health & Welfare, Republic of Korea (grant number: HI15C0395) and by research fund of Chungnam National University. This work was also supported by the Research Program on Emerging and Re-emerging Infectious Diseases (JP18fk0108047 to M.Y.) and Japanese Initiative for Progress of Research on Infectious Diseases for global Epidemic (JP18fm0208018 to M.Y.) from Agency for Medical Research and Development (AMED).

## Author contributions

J.K.K. and Y.S.K. performed the experiments including mouse work, and contributed equally to this work; H.-M.L. performed CFU assay in MDMs and fly work; Y.S.K., H.S.J., and S.-K.C. performed zebrafish work and analyzed the data; S.-H.L. determined transmission electron microscopy images; C.N. and J.B.P. performed voltage-clamp recordings; J.-J.M., J.-M.K., J.-H.J., H.E.C., M.S., M.Y., J.B.P., and E.-K.J. supported analytical tools, analyzed data, and provided the reagents and mice; J.K.K., Y.S.K., S.K., J. B.P., and E.-K.J. analyzed and discussed the results; E.-K.J. and J.B.P. designed and supervised the study, wrote the manuscript.

## Additional information

**Competing interests:** The authors declare no competing interests.

