## [Peer Review File · Nature Communications]

Reviewers' comments:

Reviewer #1 (Remarks to the Author):

The present study by Kim and colleagues investigates the role of GABAergic stimulation on host defense against intracellular microorganisms. The authors show that GABA stimulation stimulates autophagy and host defense mechanisms, while blockade of the pathway results in increased susceptibility. The concept presented here is novel and relevant, and the authors have presented a clear argumentation for the importance of the study. The experiments are well performed and the study is clearly written.

Comments:

1. The experiments look well performed and in line with the conclusions of the authors, but it would be good to discuss one set of data that may be puzzling in this respect. The concept would be that GABA induces autophagy, and this is important for host defense. However, the authors show that BCG infection of the cells reduces GABA, while numerous data in the literature have shown that BCG induces autophagy. How can these be reconciled?
2. While GABA itself does not induce cytokine gene transcription, as shown by the authors, what is the effect of GSBA on MTB/bacterial-induced cytokine production?
3. The authors should be commended for the extensive validations in more in-vivo systems, which gives the confidence that the process observed is truly solid. While it is important that the authors mention that the data shown are at least from two duplicate experiments, the number of animals per experiment should also be given.
4. Can the authors also speculate about possible effect of GABA on lymphocytes as well? This discussion is important for interpreting the in-vivo studies.

Reviewer #2 (Remarks to the Author):

Summary: This manuscript examines GABAergic signaling in host protection against intracellular pathogens, including *Mycobacterium tuberculosis*. GABA is an inhibitory neurotransmitter that is typically studied in context of primary neurologic and behavioral diseases. Here, the investigators use many different methods to study role of GABA to influence immune and inflammatory responses. They determined that GABAergic activation of innate immune cells (macrophages primarily) mediated antimicrobial responses by the GABA-A-Receptor on macrophages and induced autophagy. In contrast, GABAergic inhibition had the opposite effect.

Strengths of the manuscript lie in the novelty of examining what is typically studied in context of the nervous system and the multiple methods employed to determine the underlying mechanisms. In vivo and in vitro studies were conducted, using multiple pathogens, multiple animal models, and some human-sourced materials. An additional strength is that the style is clear and concise.

Minor weaknesses are in the results presentation and analysis:

1. Figures 2D and 7B, examples of photomicrographs are shown but the quantification of lesions is not. From the methods it appears that slides were digitally scanned and then quantification was performed. Is it possible to include that quantification?
2. Line 291 What is meant by "kindling"
3. Lines 298/299/300 Please consider including the data about depleting macrophages.
4. In the discussion please consider discussing some of the limitations of the studies, for example for some of the mouse experiments IV infection may not well-represent aerosol infection.
5. If possible, extending the experimental Mtb infection for longer than 2 weeks may be very informative, for example to determine whether GABAergic influence is most important during early infection, or has influence when acquired immunity becomes established, or during chronic

infection. It would be a nice complement to the Salmonella survival studies.

Reviewer #3 (Remarks to the Author):

The manuscript by Kim et al studies effects of GABA and drugs that affect the GABAA receptors on macrophages from mice and humans. The authors claim vital role of GABA and GABAA receptors in innate immune responses against bacterial infections where the effects of GABA on autophagy are particularly important. The effect of GABA and other neurotransmitters on immune cells is a relatively new but growing field of research. While some of the observations in the current manuscript are clearly interesting there are significant shortcomings of the manuscript in its current form.

Major problems.

1.

a. Important proper controls lacking, Fig 1a, GABAA receptors respond immediately to GABA and not with a delay as the figures show. As the time scale is lacking for the x-axis it is impossible to judge just how slow this response is! To prove that this response is indeed due to opening of GABAA receptors, the inhibitors should be applied during the GABA application to see if they can stop the response. In this experiment, the control was pre-application of picrotoxin (PTX). PTX is an open channel blocker and thus needs GABA to first open the channels before it is applied. Thus, pre-application of PTX and then GABA does not work as a control – and in any case, the data should be shown for the control experiments and that was not done here. Fig 5e, if this is a true response to GABA then a control of GABA together with an antagonist should be shown were the response is blocked.

b. Fig 1b, GABA antibody, no control experiments are provided proving the specificity of the antibody. Since this experiment is crucial to the paper demonstrating the specificity of the antibody is of uttermost importance.

c. Fig 1g, 3c, 5a what are the units on the X-axis

d. Fig 1f, here the authors show that the relevant/physiological GABA concentration is 1-2 microM but nevertheless the authors use mostly 100 microM (Fig 2, Fig 3, Fig 4, Fig 5, Fig 6) even 1 mM (Fig 1a) for most of the experiments in the manuscript. One has to raise the question, what is the relevance of these experiments if the GABA conc used is never experienced in vivo and is probably way too high to have in the bloodstream as at this concentration it probably would be able to cross the blood brain barrier by unspecific mechanisms!

e. Fig 1i and j, what were the reference genes here? Why were not other subunits for the GABAA receptors examined? What were there relative expression levels of these three genes examined? Were they expressed at the same or different level? This matters when it comes to forming functional receptors.

f. Fig 5g, the authors seem to think that incubating cells with BAPTA will decrease intracellular Ca²⁺. BAPTA is usually used as an intracellular chelator for Ca²⁺ and is only effective if it has access to the intracellular milieu.

g. When one uses drugs, it is essential to give the specific concentration of the drugs in the figures as receptors have specific affinities and other properties that are directly related to drug concentrations. The authors use shades of black to indicate concentrations in many Figs e.g. Fig 2g, Fig 5g, 5i, Fig 6g, and this is inappropriate.

h. Statistics, was the best statistical test used best for the experiments? see f.ex. Fig 2a, 6a and so on. But even more importantly – the last sentence starting "Data are means of at least duplicates from two experiments", this sentence is repeated in the figure legends for all 7 figures! What does this mean, only data from two animals used? or that the experiment is repeated twice or done twice? Are the experiments technical repeats or biological repeats? This statement is of major concern for all the different types of experiments shown in the manuscript.

2. In methods, abstract and discussion, the authors indicate that they use mice and zebrafish. By

carefully reading the Figure legends I am unable to find the zebrafish data. Further in the authors contributions people are thanked for their contributions including experiments on flies! I am unable to find any fly work in this manuscript.

3. The discussion is difficult to follow, is partly repetitive of the introduction and includes statements that are not supported by the experimental data shown.

4. The English of the manuscripts needs to be improved.

Reviewer #1 (Remarks to the Author):

The present study by Kim and colleagues investigates the role of GABAergic stimulation on host defense against intracellular microorganisms. The authors show that GABA stimulation stimulates autophagy and host defense mechanisms, while blockade of the pathway results in increased susceptibility. The concept presented here is novel and relevant, and the authors have presented a clear argumentation for the importance of the study. The experiments are well performed and the study is clearly written.

Response:

Thanks for your kind and excellent comments. We submit a revised version of our manuscript and a point-by-point response to the reviewers' comments. Detailed responses are described below.

Comments:

1. The experiments look well performed and in line with the conclusions of the authors, but it would be good to discuss one set of data that may be puzzling in this respect. The concept would be that GABA induces autophagy, and this is important for host defense. However, the authors show that BCG infection of the cells reduces GABA, while numerous data in the literature have shown that BCG induces autophagy. How can these be reconciled?

Response:

Why does BCG infection reduce GABA in macrophages, even though it induces autophagy? To our knowledge, BCG infection *per se* is not a good inducer of xenophagy in murine bone marrow-derived macrophages (BMDMs) since it has no ESX-1 T7SS (Watson RO et al., (2012) Cell. 150(4):803-815). In RAW264.7 cells, BCG infection *per se* did not significantly affect the conversion of LC3-I to II (Fig. 3 in Guo L et al., (2016) PLoS One. 12(6):e0179772). In human primary monocytes, BCG stimulation increases the amount of LC3+ vesicles (Fig. 3 in Buffen K et al., (2014) PLoS Pathog. 10(10):e1004485). Thus, BCG can induce autophagy depending on the cell type.

Rather than a direct inducer of autophagy, *M. bovis* BCG bacilli can be restricted by autophagy pathway activation triggered by numerous exogenous stimuli through diverse

mechanisms. For example, interferon (IFN)- γ activates autophagy to restrict BCG replication in macrophages via interferon-induced GTPases (Gutierrez MG et al., (2004) Cell. 119(6):753-766). Recent studies have shown that calcimycin is an inhibitor of BCG growth through autophagy induction via intracellular calcium-dependent adenosine triphosphate (ATP) release (Mawatwal S et al., (2017) Biochim. Biophys. Acta. 1861(12):3190-3200). In this study, GABAergic activation increased autophagy to promote phagosomal maturation and host defenses against infection with BCG and Mtb through a mechanism involving the Ca^{2+} -AMPK-GABARAPL1 pathway.

Indeed, numerous intracellular bacteria have evolved diverse strategies to modulate, suppress, or hijack the xenophagy system to escape from host innate defenses (Bah A et al., (2017) Front. Immunol. 8:1483; Huang J et al., (2014) Nat. Rev. Microbiol. 12(2):101-114). Therefore, the activation of autophagy by treatment with various small molecules or reagents may contribute to the development of host-directed therapies against intracellular pathogens.

Because of the limitation of word counts in this study, we could not include the Discussion in the revised manuscript. However, a part of these comments have been included in Discussion (page 17-18, lines 435-437), as follows:

“Recently, increasing efforts have been made to develop several autophagy-targeting agents as potential host-directed therapeutics to overcome drug resistance issues in a variety of infectious diseases, including TB⁵³⁻⁵⁵.”

2. While GABA itself does not induce cytokine gene transcription, as shown by the authors, what is the effect of GSBA on MTB/bacterial-induced cytokine production?

Response:

You raise an interesting point. We performed a quantitative reverse transcription polymerase chain reaction (qRT-PCR) analysis of lung samples from Mtb-infected mice treated with/without GABA (same conditions used in Fig. 2a). The *Tnf* and *Il6* mRNA levels were significantly decreased in the lung samples by GABA treatment compared with those from untreated/Mtb-infected controls (Supplementary Fig.4a, b).

We also examined GABA-mediated cytokine expression in Mtb-infected BMDMs. Mtb-infected BMDMs were treated with or without GABA, and the mRNA expression of proinflammatory cytokines tumor necrosis factor alpha (TNF- α) and interleukin 6 (IL-6) was

measured. As shown in Supplementary Fig.4c and d, in BMDMs, GABA treatment led to a significant reduction in *Tnf* only at the highest dose of GABA, whereas it did not affect *Il6* mRNA expression. Together, GABA treatment led to the inhibition of inflammatory cytokine generation *in vivo* during *Mtb* infection. However, in macrophages, the effects of GABA may be minimal in the regulation of proinflammatory cytokine production.

These data have been included in Supplementary Fig.4a-d, the Results (page 7), and the Discussion (page 15), as follows:

In Results (page 7, lines 151-159), as follows:

“We then examined whether GABA treatment led to significant inhibition of Tnf and Il6 in lung tissues and BMDMs during Mtb infection. We found that Tnf and Il6 mRNA levels were significantly decreased in the infected lung samples by GABA treatment, when compared with those from untreated/Mtb-infected controls (Supplementary Fig. 4a, b). In BMDMs, GABA treatment led to a significant reduction of Tnf only at the highest dose of GABA, whereas it did not affect Il6 mRNA expression at any dose of GABA (Supplementary Fig. 4c, d). Taking these findings together, GABA treatment led to the inhibition of inflammatory cytokine generation in vivo during Mtb infection; however, in macrophages, GABA had a minimal effect on proinflammatory cytokine production.”

In Discussion (page 15, lines 360-363), as follows:

“In addition, GABA treatment ameliorated the inflammatory cytokine production in the infected sites (lungs) during Mtb infection. Pathologic inflammation and neutrophil infiltration, as well as excessive tumor necrosis factor production, are deleterious in mice and zebrafish models with tuberculosis (TB)³⁵⁻³⁸.”

3. The authors should be commended for the extensive validations in more in-vivo systems, which gives the confidence that the process observed is truly solid. While it is important that the authors mention that the data shown are at least from two duplicate experiments, the number of animals per experiment should also be given.

Response:

We apologize for our incorrect description in the figure legends. In our revision, we have corrected our description of the exact number of animals and representative experiments in detail throughout the paper.

4. Can the authors also speculate about possible effect of GABA on lymphocytes as well? This discussion is important for interpreting the in-vivo studies.

Response:

GABAergic activation may play multiple roles in a variety of immune cells, including lymphocytes. It was reported that lymphocytes possess components of the GABAergic system, including the GABA synthesis enzyme glutamic acid decarboxylase, GABA transporter (GAT)-1 and GAT-2, as well as a number of type A GABA receptor (GABA_AR) subunits (Barragan A et al., (2015) *Acta. Physiol. (Oxf)*. 213(4):819-827).

Although the molecular mechanisms have not been fully recognized, several previous studies have shown the regulatory effect of GABA on lymphocytes. Earlier studies reported that either GABA or a GABA_AR antagonist inhibits T cell proliferative responses to anti-CD3 or a specific antigen, and that GABA suppresses delayed-type hypersensitivity responses *in vivo* (Tian J et al., (1999) *J. Neuroimmunol.* 96(1):21-28). The same group further showed that a relatively low dose of GABA (600 µg daily) had a beneficial effect by inhibiting proinflammatory Th1 responses to β cell antigens and suppressed disease progression in prediabetic NOD mice (Tian J et al., (2004) *J. Immunol.* 173(8):5298-5304). In addition, the stimulation of human T cells with anti-CD3 monoclonal antibodies (mAbs) or concanavalin A exerted inhibitory effects on T cell proliferation in a GABA_AR-dependent manner (Prud'homme GJ et al., *Transplantation.* (2013) 96(7):616-623). A recent study showed that the GABA_AR α4 subunit plays a role in the modulation of lung T cell activation, CD4⁺ T cell proliferation, and cytokine production during airway inflammation (Yocum GT et al., *Am. J. Physiol. Lung Cell Mol. Physiol.* 313(2):L406-L415). Furthermore, treatment with honokiol, a phenolic GABA_AR agonist, led to a marked inhibitory effect on Th1- and Th17-type inflammatory cytokines during lung inflammation (Munroe ME et al., (2010) *J. Immunol.* 185(9):5586-5597).

GABA treatment also increased the suppressive effect of rapamycin on lymphocyte proliferation in humans and mice (Prud'homme GJ et al., (2013) 96(7):616-623). Furthermore, GABA treatment augmented the secretion of the inhibitory cytokine transforming growth factor beta 1 (TGF-β1) and regulatory T cell numbers (Soltani N., (2011). *Proc. Natl. Acad. Sci. U S A.* 108(28):11692-11697). Taken together, GABAergic activation may promote an inhibitory effect on inflammatory Th1- and Th-17 lymphocytes and simultaneously induce immunoregulatory molecules.

GAT-1 is critically involved in the maintenance of GABA and functions in diverse cells/tissues (Barragan A et al., (2015) *Acta. Physiol. (Oxf)*. 213(4):819-827; Ren W et al., (2017) *Cell Death Dis.* 8(3):e2655; Egawa and Fukuda (2013) *Front. Neural. Circuits.* 7:170). Previous studies have suggested a negative regulatory role for GAT-1 in the differentiation of Th1 and Th17 cells (Ren W et al., (2017) *Cell Death Dis.* 8(3):e2655). GAT-1 also exerts an inhibitory function in CD4⁺ T cell cycle entry from G₁ to S phase, thus affecting cell division (Wang Y et al., (2008) *J. Immunol.* 183(5):3488-3495). In addition, a GAT-1 deficiency enhances nuclear factor kappa B (NF-κB) and the T-bet-STAT1 signaling pathway, as well as IFN-γ-producing Th1 cell responses, resulting in the aggravation of experimental autoimmune encephalomyelitis pathogenesis (Wang Y et al., (2008) *J. Immunol.* 183(5):3488-3495). Thus, GAT-1 and the maintenance of GABA levels may be important for immune homeostasis.

Although we did not examine the effects of GABA on lymphocyte activation during a mycobacterial or *Salmonella* infection, we speculate that GABA treatment decreases inflammatory Th1 cells and the production of IFN-γ, but increases regulatory T cell responses. IL-12/IFN-γ-mediated Th1 immunity is a critical immune defense against Mtb infection (O'Garra A et al., (2013) *Annu. Rev. Immunol.* 31:475-527). However, excessive stimulation of Th1 immunity in tuberculosis (TB) results in pathologic inflammation (Mourik BC et al., (2017) *Front. Immunol.* 8:294). In addition, type I interferons (T1-IFNs) and Th17 immunity are involved in TB pathogenesis (Lyadova IV et al., (2015) *Mediators Inflamm.* 854507; McNab F et al., (2015) 15(2):87-103). Together, the maintenance of GABA levels may help regulate protective immune responses and avoid pathologic inflammation during TB infection. Future studies should clarify the exact function of GABAergic activation upon adaptive immune responses during intracellular bacterial infections.

These comments have been included in the Discussion section (page 15-16, lines 364-377), as follows:

“Additionally, earlier studies indicated the inhibitory effects of GABA on T-cell proliferation, proinflammatory Th1 responses, and delayed-type hypersensitivity (DTH) responses in vivo^{39, 40}. Moreover, treatment of honokiol (HNK), a phenolic GABA_AR agonist, led to a marked inhibitory effect on Th1- and Th17-type inflammatory cytokines during lung inflammation⁴¹. Although we did not examine the effects of GABA on lymphocyte activation in mycobacterial or salmonella infection, we speculate that GABA treatment decreases inflammatory Th1 cells and regulates proinflammatory T-cell responses. Excessive stimulation of Th1 immunity in TB can result in pathologic inflammation⁴². In addition, type I interferons (T1-IFNs) and Th17 immunity have been

recognized to be involved in TB pathogenesis^{43, 44}. Combined with the current data, GABAergic activation may contribute to host defenses and simultaneously control excessive inflammation associated with pathologic responses during bacterial infection. Future studies should be performed to clarify the exact function of GABAergic activation upon adaptive immune responses during intracellular bacterial infection.”

Reviewer #2 (Remarks to the Author):

Summary: This manuscript examines GABAergic signaling in host protection against intracellular pathogens, including *Mycobacterium tuberculosis*. GABA is an inhibitory neurotransmitter that is typically studied in context of primary neurologic and behavioral diseases. Here, the investigators use many different methods to study role of GABA to influence immune and inflammatory responses. They determined that GABAergic activation of innate immune cells (macrophages primarily) mediated antimicrobial responses by the GABA-A-Receptor on macrophages and induced autophagy. In contrast, GABAergic inhibition had the opposite effect.

Strengths of the manuscript lie in the novelty of examining what is typically studied in context of the nervous system and the multiple methods employed to determine the underlying mechanisms. In vivo and in vitro studies were conducted, using multiple pathogens, multiple animal models, and some human-sourced materials. An additional strength is that the style is clear and concise.

Response:

Thanks for your kind and excellent comments. We submit a revised version of our manuscript and a point-by-point response to the reviewers' comments. Detailed responses are described below.

Minor weaknesses are in the results presentation and analysis:

1. Figures 2D and 7B, examples of photomicrographs are shown but the quantification of lesions is not. From the methods it appears that slides were digitally scanned and then quantification was performed. Is it possible to include that quantification?

Response:

We have included the quantification data for Fig. 2d and Fig. 7b. The histopathological score was assigned to lung tissue from each animal based on the extent of granulomatous inflammation as follows: 0 = no lesion, 1 = minimal lesion (1–10% of tissue in section involved), 2 = mild lesion (11–30% involved), 3 = moderate lesion (30–50% involved), 4 = marked lesion (50–80% involved), and 5 = severe lesion (.80% involved), as described previously (Sweeney, K. A. et al., (2011). *Nat. Med.* 17: 1261–1268). The histopathological score was graded for severity by scanning at least 20 random fields in 3 sections per mouse.

These results have been included in right panels of Fig. 2d and Fig. 7b, and their figure legends. In addition, the description has been included in Materials and Methods (page 27-28, lines 682-688), as follows:

“The histopathological score was graded for severity by scanning multiple random fields in three sections per mouse. An overall histopathological score was assigned to lung tissue in each animal based on the extent of granulomatous inflammation as follows: 0 = no lesion, 1 = minimal lesion (1%–10% of tissue in section involved), 2 = mild lesion (11%–30% involved), 3 = moderate lesion (30%–50% involved), 4 = marked lesion (50%–80% involved), and 5 = severe lesion (>80% involved), as described previously⁶³.”

2. Line 291 What is meant by "kindling"

Response:

Kindling is one of the most widely used models of seizures and epilepsy to provide key insight into seizures and epilepsy. It refers to a seizure-induced plasticity phenomenon that occurs when repeated seizure induction by electrical or chemical stimulation evokes the progressive enhancement of seizure susceptibility.

In response to this comment, we have rephrased “epilepsy and kindling” with “epilepsy and kindling (progressive enhancement of seizure susceptibility)”. Given that GABA_AR antagonists caused no progressive changes in seizure susceptibility (kindling), the authors believe that the antagonists increased the overall susceptibility to infection and bacterial loads with minimal involvement of the central nervous system (CNS).

The comments have been included in Discussion (page 14, line 322-325), as follows:

“Our data demonstrate that GABAergic activation promotes antimicrobial host defenses, whereas GABAergic inhibition (under conditions that did not cause epilepsy and kindling; progressive enhancement of seizure susceptibility) increased the overall susceptibility to infection and bacterial loads both in vitro and in vivo.”

3. Lines 298/299/300 Please consider including the data about depleting macrophages.

Response:

We have included the macrophage depletion data in Supplementary Fig.5. In our preliminary experiments, we evaluated whether GABA treatment plays a critical role in macrophage-mediated innate immune defenses. Using clodronate liposomes, we found that the protective effects of GABA on the survival of mice infected with *Salmonella* were almost completely abrogated (Supplementary Fig.5a). In addition, clodronate-liposome treatment caused equal bacterial growth in both GABA-treated and untreated control groups (Supplementary Fig.5b-c). Overall, these data suggest that macrophages play a critical role in GABAergic protective responses against intracellular bacterial infections.

These data have been included in Supplementary Fig.5; Results and Discussion, as follows:

In Results (page 7-8, lines 166-175), as follows:

“In addition, we examined whether GABA treatment played a critical role for innate immune defense in macrophages. When clodronate liposomes were administered, the GABAergic protective effects upon mouse survival in S. typhimurium infection were completely abrogated (Supplementary Fig. 5a). In addition, clodronate-liposome treatment resulted in equal bacterial growth in both GABA-treated and untreated control mice (Supplementary Fig. 5b and c, for S. typhimurinum and BCG, respectively). Together, these results clearly show that GABAergic activation triggered the antimicrobial host defense and ameliorated pathologic inflammation against intracellular bacterial infections. In addition, macrophages may play a critical role in GABAergic protective responses against intracellular bacterial infection.”

In Discussion (page 14, lines 327-330), as follows:

“Herein, we focused on macrophages, the principal immune cells to combat intracellular bacterial infection, in GABAergic host defense. Importantly, macrophage depletion significantly affected the GABAergic defense system during infections.”

4. In the discussion please consider discussing some of the limitations of the studies, for example for some of the mouse experiments IV infection may not well-represent aerosol infection.

Response:

We agree with your points. Although numerous previous studies have conducted *in vivo* Mtb/BCG experiments via the intravenous infection route, particularly for evaluating chemotherapeutic or vaccine effects against TB (see the references below), this does not represent aerosol infection.

Similar to intravenous infection, we found that GABA treatment promoted host defenses in mouse models with intranasal BCG infection (Figs. 2c; 7a, right; and 7c). We have included these comments as experimental limitations in the Discussion (page 15, lines 357-360), as follows:

“Although there are some experimental limitations in using an intravenous infection model, which may not represent aerosol infection well, we found that GABA treatment suppressed in vivo mycobacterial loads in intranasal and intravenous infection models.”

References in which intravenous infection was introduced)

- Pan H, Nature. 2005 Apr 7;434(7034):767-72. PMID: 15815631 (10⁵ live *M. tuberculosis*)
- Roy E, Infect Immun. 2005 Sep;73(9):6101-9. PMID: 16113331 (2 x 10⁵ live *M. tuberculosis*)
- Sweeney KA, Nat Med. 2011 Sep 4;17(10):1261-8. doi: 10.1038/nm.2420. PMID: 21892180 (1 × 10⁷ CFU Mtb H37Rv per mouse)
- Festjens N, EMBO Mol Med. 2011 Apr;3(4):222-34. PMID: 21328541 (5 x 10⁴ Mtb H37Rv)
- Agarwal P, Microbes Infect. 2014 Jul;16(7):571-80. PMID: 24819214 (8 x 10⁴ or 8 x 10⁵ Mtb H37Rv)
- Roach DR, Infect Immun. 1999 Oct;67(10):5473-6. PMID: 10496932 (10⁶ *M. bovis* BCG)
- Chackerian AA, Infect Immun. 2006 Nov;74(11):6092-9. Epub 2006 Aug 21. PMID: 16923792 (2.3 x 10⁵ to 5 x 10⁶ CFU of BCG *M. bovis* BCG)

5. If possible, extending the experimental Mtb infection for longer than 2 weeks may be very informative, for example to determine whether GABAergic influence is most important during early infection, or has influence when acquired immunity becomes established, or during chronic infection. It would be a nice complement to the Salmonella survival studies.

Response:

Regarding the mouse experiments using the Mtb strain, we were unable to use the ABL3 facility until the end of August. Therefore, we performed the experiments using *M. bovis* BCG for 3 weeks after infection and found no significant differences in the bacterial loads *in vivo*

after 3 weeks of infection (see Fig. 1 for review purposes only). These data strongly suggest that a GABAergic defense is principally required for protection during early infection with mycobacteria. As noted by the reviewer, these data seem to be in agreement with the importance of GABAergic protection against *Salmonella* infection.

However, we believe these data should be examined under Mtb-infected conditions in the future. Therefore, we ask that the reviewer allow us to clarify this issue in a future study.

Reviewer #3 (Remarks to the Author):

The manuscript by Kim et al studies effects of GABA and drugs that affect the GABAA receptors on macrophages from mice and humans. The authors claim vital role of GABA and GABAA receptors in innate immune responses against bacterial infections where the effects of GABA on autophagy are particularly important. The effect of GABA and other neurotransmitters on immune cells is a relatively new but growing field of research. While some of the observations in the current manuscript are clearly interesting there are significant shortcomings of the manuscript in its current form.

Response:

Thanks for your kind and excellent comments. We submit a revised version of our manuscript and a point-by-point response to the reviewers' comments. Detailed responses are described below.

Major problems.

1. a. Important proper controls lacking, Fig 1a, GABAA receptors respond immediately to GABA and not with a delay as the figures show. As the time scale is lacking for the x-axis it is impossible to judge just how slow this response is!

Response:

We have included proper labels for the time scales of each current trace in our revised manuscript (see Fig. 1a).

With the time scales, it is now clear that in-bath GABA application caused an immediate inward shift in the holding current.

To prove that this response is indeed due to opening of GABAA receptors, the inhibitors should be applied during the GABA application to see if they can stop the response. In this experiment, the control was pre-application of picrotoxin (PTX). PTX is an open channel blocker and thus needs GABA to first open the channels before it is applied. Thus, pre-application of PTX and then GABA does not work as a control – and in any case, the data should be shown for the control experiments and that was not done here.

Response:

The authors understand the reviewer's concern that pretreatment with picrotoxin is not the first choice to prevent GABA_AR activation. In response, we performed a new set of experiments. We examined whether GABA_AR antagonists block GABA-induced transient and tonic currents before and during GABA application; the results have been added to Fig. 1a in the revised manuscript.

As shown in Fig. 1a in the revised manuscript, both transient and tonic GABA-induced currents were efficiently blocked by bicuculline (BIC). Similar results were observed with picrotoxin, as described in the original manuscript under our recording conditions.

In addition, these comments have been included in Results (page 5, lines 96-111), as follows:

“To assess whether the GABAergic system in bone marrow-derived macrophages (BMDMs) is modulated during infection, we first used voltage-clamp recording to confirm that BMDMs express functional GABA receptors (Fig. 1a). In the whole-cell patch-clamp mode, in-bath application of GABA to BMDMs evoked transient inward currents in all tested cells (Fig. 1a). In the presence of bicuculline (BIC), a GABA_AR antagonist, GABA failed to cause an inward current (n = 5), suggesting that GABA evoked currents predominantly by binding to GABA_ARs. GABA-induced currents were composed of the initial peak current and residual long-lasting current; the latter was more significant at a higher concentration of GABA. Thus, we investigated whether GABA could activate tonic currents in BMDMs. GABA applied in the perfusion solution (flow rates, ~3 ml/min) evoked tonic currents of lower amplitude (ranging from -15 to -39 pA) in 7 of 12 BMDMs, while it failed to generate a tonic current in all seven tested cells in the presence of BIC. Tonic currents were also efficiently blocked by the additional application of BIC (Fig. 1a, inset). Similar results were observed with another GABA_AR antagonist, picrotoxin (PTX; data not shown). These results suggest that BMDMs express functional GABA_ARs responding with transient or tonic currents when exposed to GABA.”

Fig 5e, if this is a true response to GABA then a control of GABA together with an antagonist should be shown were the response is blocked.

Response:

The reviewer is correct, in that we missed an important control experiment. In response, we performed a new set of experiments; the results have been added to Fig. 5e in the revised manuscript. We investigated whether pre-incubation with a selective GABA_AR antagonist, BIC, blocked the increase in intracellular Ca²⁺ concentration ([Ca²⁺]_i) induced by GABA application.

As shown in Fig. 5e in the revised manuscript, BIC efficiently blocked the [Ca²⁺]_i increase induced by GABA, but not by ATP. These results confirm that GABA-induced intracellular Ca²⁺ release is mediated via GABA_AR activation.

These data have been included in Fig. 1a and 5e, Results (Page 11, lines 256-259), as follows:

"It was also noted that pre-incubation of a selective GABA_AR antagonist, BIC, efficiently blocked [Ca²⁺]_i increase induced by GABA but not by ATP (Fig. 5e, right), indicating that GABA-induced intracellular calcium release is mediated through GABA_AR activation."

b. Fig 1b, GABA antibody, no control experiments are provided proving the specificity of the antibody. Since this experiment is crucial to the paper demonstrating the specificity of the antibody is of uttermost importance.

Response:

Although we examined the specificity of the primary antibody (Ab) (anti-GABA Abs) in our preliminary study, we clarified the specificity of the anti-GABA Ab using three methods: 1) using an isotype control Ab to determine the specificity of primary Ab binding to the antigen GABA; 2) using a secondary Ab control that shows the label is specific to the primary Ab; and 3) using another anti-GABA mAb and its control Ab.

1) The level of fluorescence resulting from staining with the isotype control Ab (normal rabbit IgG; sc-2027) was undetected, demonstrating that GABA staining with the anti-GABA Ab (rabbit polyclonal; A2052) was highly specific (Supplementary Fig. 1a).

2) We eliminated the primary Ab and incubated the membrane with the secondary Ab (Alexa Fluor 488-conjugated anti-rabbit IgG; A17041) alone. No signal was detected (Supplementary Fig. 1b).

3) We also confirmed our GABA imaging results using another anti-GABA Ab (mouse IgG1 monoclonal; ab86186) and its isotype-matched control (mouse IgG1; ab81032) (see Fig. 2 for review purposes only). The control and alternate primary Abs confirmed the specific binding of the primary Abs for GABA staining.

c. Fig 1g, 3c, 5a what are the units on the X-axis

Response:

The units of horizontal axis are the fluorescence (FITC) intensity against the number of events detected (vertical axis).

We have included these comments in Figure legends of Figs 1g (lines 1006-1007), 3c (lines 1047-1048), and 5a (lines 1094-1095).

d. Fig 1f, here the authors show that the relevant/physiological GABA concentration is 1-2 microM but nevertheless the authors use mostly 100 microM (Fig 2, Fig 3, Fig 4, Fig 5, Fig 6) even 1 mM (Fig 1a) for most of the experiments in the manuscript. One has to raise the question, what is the relevance of these experiments if the GABA conc used is never experienced *in vivo* and is probably way too high to have in the bloodstream as at this concentration it probably would be able to cross the blood brain barrier by unspecific mechanisms!

Response:

Regarding the physiological relevance of GABA, please refer to the following: 1) Fig. 7, 2) the GABA concentration used *in vivo* in previous studies, and 3) behavioral tests.

1) Fig. 7

In Fig. 7, all mouse/macrophage experiments were performed without GABA treatment *per se*. When the macrophages were treated with antagonists of GABA_AR, the intracellular or *in vivo* bacterial loads were significantly increased. These data strongly suggest that the basal levels of GABA and functional GABA_AR signaling are critically involved in the appropriate induction of host defenses against an intracellular bacterial infection.

2) The GABA concentration used in previous *in vivo* studies

The concentration range of GABA used in this study (100 μ M to 1 mM) was referred from numerous previous reports (see Table 1 below for review purposes only) that investigated the function of GABA in the regulation of peripheral immune responses.

Table 1 for the review purpose only. GABA dose used *in vivo/in vitro* in previous literatures

In vivo, GABA administration			
	GABA	Route	Reference
Human	100mg	Orally	Abdou, A.M., et al., Biofactors. 2006;26(3):201-8
	800 mg	Orally	Steenbergen, L., et al., Sci Rep. 2015 Jul 31;5:12770.
	0.8g/kg	Orally	D. B. Tower, in "Inhibition in the Nervous System and Gamma-Aminobutyric Acid," p. 562-578. Pergamon Press, Oxford, 1960
Rat	100 mg/100 g body weight	Orally	Abdou, A.M., et al., Biofactors. 2006;26(3):201-8
	100 or 500 mg/kg	Orally	Kim, H.Y., et al., Food Chem Toxicol. 2004 Dec;42(12):2009-14.
	10 mg/kg, 100-500 mg/kg	intraparitoneal injection	Borycz, J., et al., J Physiol Pharmacol. 1992 Sep;43(3):259-69.
Mice	10 mg/kg	intraparitoneal injection	http://digitalcommons.cwu.edu/source/2014/posters/151/
	2-2000 mg/kg (mainly 200mg/kg)	intraparitoneal injection	Carmans S et al., J Neuroimmunol. 2013 Feb 15;255(1-2):45-53.
	25-2000 mg/kg (mainly 100 mg/kg)	intraparitoneal injection	Biswas, B., et al., Psychopharmacology (Berl). 1978 Sep 15;59(1):91-4.
	0.04-40 mg/kg	intraparitoneal injection	Ren W et al., Front Immunol. 2017 Jan 16;7:685.
In vitro, GABA treatment			
	Cell type	Concentration	Reference
Mice	Peritoneal macrophage	GABA 0.1-1 mM	Carmans S et al., J Neuroimmunol. 2013 Feb 15;255(1-2):45-53.
	Splenic mononuclear cells	GABA 0.1-3 mM	Tian J et al., J Immunol. 2004 Oct 15;173(8):5298-304.
	Dendritic cells	GABA 0.5 μ M, Muscimol 300 μ M, Bicuculline 50 μ M	Fuks, J.M. et al., PLoS Pathog. 2012;8(12):e1003051.
	Peritoneal macrophages	Muscimol 10–100 μ M	Bhat R et al., Proc Natl Acad Sci U S A. 2010 Feb 9;107(6):2580-5.
	Dendritic cells	GABA 1 μ M or 1 mM	Fuks, J.M. et al., PLoS Pathog. 2012;8(12):e1003051.
Human	Dendritic cells	GABA 1 μ M or 1 mM	Fuks, J.M. et al., PLoS Pathog. 2012;8(12):e1003051.

3) Behavioral tests

Finally, we examined whether the maximal doses of GABA used in our study led to any undesirable effects on the neurological system.

As shown in Fig. 3 (for review purposes only), GABA injection (200 mg/kg, i.p., q.i.d.) caused no significant behavioral changes, as determined by the open field test (Fig. 3a,b for review purposes only), the rotarod test (Fig. 3c for review purposes only), and seizure sensitivity to electroshock (Fig. 3d for review purposes only). This interpretation is consistent with results showing that a GABA injection (200–400 mg/kg, i.p., q.i.d.) did not increase the brain GABA concentration, while the higher concentration of GABA (600 mg/kg) increased brain GABA (Shyamaladevi N et al., (2002) Brain Res. Bull. 57(2):231-236).

Combined with the fact that GABA_AR antagonists themselves significantly increased the intracellular or *in vivo* bacterial loads (Fig. 7), the additional data support the notion that peripheral GABA and GABA_AR signaling in macrophages is critically involved in host defenses against intracellular bacterial infections with minimal involvement of the CNS. The authors politely argue against the concern that the GABA concentrations used in the current study may affect brain GABA concentrations.

However, the authors fully agree that we cannot exclude the possibility that plasma GABA, especially at the higher concentration, crosses the blood brain barrier (BBB) because it is not possible at this time to draw a definite conclusion with regards to the BBB permeability of GABA. A number of studies were unable to show that GABA crosses the BBB, whereas several other studies showed GABA's ability to cross. In a recent review (Boonstra E et al., (2015) Front. Psychol. 6:1520), this discrepancy was fully discussed; it involves variations in the chemical compounds, method of administration, and species used. It is important to note that the review suggested that GABA supplements may work through the enteric nervous system rather than the CNS, although additional research is required to support this hypothesis.

These comment have been included in Discussion (lines 425-432), as follows:

“Since the experiments using antagonists of GABA_AR were performed in the absence of GABA treatment per se, the basal levels of GABA and functional GABA_AR signaling are critically involved in the appropriate induction of host defenses against an intracellular bacterial infection. In addition, GABA injection (200 mg/kg, i.p., q.i.d.) caused no significant behavioral changes, as determined by the open field test, the rotarod test,

and seizure sensitivity to electroshock (data not shown), indicating that the maximal doses of GABA used in our study led to any undesirable effects on the neurological system.”

e. Fig 1i and j, what were the reference genes here? Why were not other subunits for the GABA_A receptors examined? What were there relative expression levels of these three genes examined? Were they expressed at the same or different level? This matters when it comes to forming functional receptors.

Response:

GAPDH was used as the reference gene, and the mRNA levels of each subunit were calculated using the $2^{-\Delta\Delta C_t}$ method.

During the revision process, we performed a qRT-PCR analysis for a total of 19 subunits of GABA_AR (Olsen RW et al., (2008) Pharmacol. Rev. 60(3):243–60). As shown in Fig. 1i, we found that the mRNAs of 11 (*Gabra2*, *Gabra3*, *Gabra4*, *Gabra5*, *Gabra6*, *Gabrb2*, *Gabrb3*, *Gabrg1*, *Gabrg2*, *Gabrd*, and *Gabrq*) of the 19 GABA_AR subunits were expressed in unstimulated BMDMs. We next selected five subunits (*Gabra3*, *Gabra4*, *Gabrb3*, *Gabrd*, and *Gabrq*) whose expression was relatively abundant in BMDMs. When the mRNA expression levels of the five selected subunits were examined before and after Mtb infection, we found that all subunit mRNAs were down-regulated in BMDMs in a time-dependent manner. In the case of the *Gabrq* subunit, there was a transient increase in mRNA expression after 3 h of Mtb infection followed by a decrease (data not shown). These data suggest that the mRNAs of numerous GABA_AR subunits are expressed in BMDMs and can be modulated after Mtb infection.

These data have been included in Fig. 1i and 1j, and Results (page 6, lines 124-132), as follows:

*“Furthermore, we performed real-time quantitative reverse transcription polymerase chain reaction (qRT-PCR) analysis for a total of 19 subunits of GABA_AR²⁰. As shown in Fig. 1i, the mRNAs of 11 (*Gabra2*, *Gabra3*, *Gabra4*, *Gabra5*, *Gabra6*, *Gabrb2*, *Gabrb3*, *Gabrg1*, *Gabrg2*, *Gabrd*, and *Gabrq*) among 19 GABA_AR subunits were expressed, whereas the mRNA levels of the other subunits were undetectable in resting, unstimulated BMDMs. We then selected five subunits (*Gabra3*, *Gabra4*, *Gabrb3*, *Gabrd*, and *Gabrq*) with relatively high expression in BMDMs compared with those of other subunits. When the selected five subunits levels were examined before and after Mtb*

infection, we found that all of them were down-regulated in BMDMs after infection in a time-dependent manner (Fig. 1j)."

In Discussion (page 14, lines 342-346), as follows:

"To extend these previous findings^{7, 34}, our data revealed GABA-evoked currents in primary BMDMs and confirmed the presence of 11 GABA_AR subunit transcripts in resting BMDMs. In addition, our data first demonstrate that the levels of endogenous GABA and GABA_AR subunit transcripts were decreased by infection signals."

f. Fig 5g, the authors seem to think that incubating cells with BAPTA will decrease intracellular Ca²⁺. BAPTA is usually used as an intracellular chelator for Ca²⁺ and is only effective if it has access to the intracellular milieu.

Response:

For intracellular BAPTA loading, we used 'BAPTA-AM', not 'BAPTA'. Throughout the paper, we described 'BAPTA-AM' in the text. However, we apologize for our mistake in which 'BAPTA' was used in Fig. 5g. Therefore, we have corrected 'BAPTA-AM' in Fig. 5g.

We also provide results showing that treatment with BAPTA-AM led to a decrease in the intracellular Ca²⁺ influx induced by GABA and ATP (see Fig. 4 for review purposes only).

g. When one uses drugs, it is essential to give the specific concentration of the drugs in the figures as receptors have specific affinities and other properties that are directly related to drug concentrations. The authors use shades of black to indicate concentrations in many Figs e.g. Fig 2g, Fig 5g, 5i, Fig 6g, and this is inappropriate.

Response:

We have included the specific concentration of the drugs in all figures in our revised version.

h. Statistics, was the best statistical test used best for the experiments? see f.ex. Fig 2a, 6a and so on. But even more importantly – the last sentence starting "Data are means of at least duplicates from two experiments", this sentence is repeated in the figure legends for all 7 figures! What does this mean, only data from two animals used? or that the experiment is

repeated twice or done twice? Are the experiments technical repeats or biological repeats? This statement is of major concern for all the different types of experiments shown in the manuscript.

Response:

We apologize for our incorrect description in the figure legends. In our revision, we have corrected our description of the exact number of animals and representative experiments in detail throughout the paper.

2. In methods, abstract and discussion, the authors indicate that they use mice and zebrafish. By carefully reading the Figure legends I am unable to find the zebrafish data. Further in the authors contributions people are thanked for their contributions including experiments on flies! I am unable to find any fly work in this manuscript.

Response:

We performed fly work in our original manuscript, in Supplementary Fig. 10d (in original paper, Supplementary Fig. 7c).

3. The discussion is difficult to follow, is partly repetitive of the introduction and includes statements that are not supported by the experimental data shown.

Response:

We have deleted the repetitive description and re-written the Discussion section. We believe that the revised Discussion has been improved, especially the text regarding our interpretation of the significance of our findings and key insights into our experimental data.

4. The English of the manuscripts needs to be improved.

Response:

Grammatical errors have once again been corrected by a native speaker of English.

The English in this document has been checked by at least two professional editors, both native speakers of English. For a certificate, please see:

<http://www.textcheck.com/certificate/9jDoZ9>

** See Nature Research's author and referees' website at www.nature.com/authors for information about policies, services and author benefits

This email has been sent through the Springer Nature Tracking System NY-610A-NPG&MTS

Confidentiality Statement:

This e-mail is confidential and subject to copyright. Any unauthorised use or disclosure of its contents is prohibited. If you have received this email in error please notify our Manuscript Tracking System Helpdesk team at <http://platformsupport.nature.com> .

Details of the confidentiality and pre-publicity policy may be found here <http://www.nature.com/authors/policies/confidentiality.html>

Privacy Policy | Update Profile

REVIEWERS' COMMENTS:

Reviewer #4 (Remarks to the Author):

I would like to thank the authors for including additional data to address my comments. I only have some minor comments below on the revised version.

Fig S1b, representative trace for 3 uM THIP, is BIC application missing from this trace?

Fig S2 merged images, please indicate in the figure legend that DAPI stainings are also included in the merged images.

Fig S12 bar graph, do 100 Bic+ 100 GABA groups have 4 independent experiments? The error bars were not seen.

Fig 6a, the error bar of Mtb+Iso group (Gabarapl1) was cut off.

The authors have detected only 3 GABAAR (A1, B1 and R2) in human primary MDMS, which is totally different from the results in mouse BMDMs. This may reflect the differential effect of GABA on human and mouse macrophages. The authors are advised to add some comments to address this issue.

Reviewers' comments:

Reviewer #1 (Remarks to the Author):

The authors responded appropriately to my comments.

Response:

Thanks for your kind consideration and remarks on our paper.

Reviewer #2 (Remarks to the Author):

The authors did an excellent job addressing comments. I appreciated the additional figures that were included and the helpful explanations. I have no additional concerns.

Response:

Thanks for your kind consideration and remarks on our paper.

Reviewer #4 (Remarks to the Author):

The manuscript by Kim JK et al., demonstrated the modulation of GABAA receptor-mediated GABAergic signaling in macrophage (mainly in mice) leads to the change of autophagy activation and host susceptibility to intracellular bacterial infections. The authors have shown the above findings are via the GABAergic downstream signaling molecules, e.g. intracellular Ca²⁺, AMPK, and Gabarapl1. The research question about the role of macrophage GABAergic signaling in autophagy and host innate immunity against intracellular bacterial infection is novel and is of potential interest to others especially in the field of nerve-driven immunity. The paper contains massive in vivo and in vitro data from primary cells, cell line, mouse, zebrafish, and Drosophila which appear somehow convincing. However, there are still some important questions that need further clarification from the authors to strengthen

the conclusion and hopefully could be reproduced by others. The major concerns are listed below:

Response:

Thanks for your valuable comments. We submit a revised version of our manuscript and a point-by-point response to the reviewers' comments. Detailed responses are described below.

1. fig 1a evoked GABA currents, is it true that the fast-application of GABA lasts tot the end of the trace? That the current has returned to the baseline level indicates the washout after the GABA application, so the application line should be much shorter covering only the GABA activation and desensitization current. Is the evoked GABA current dose-dependent? If so, it is reasonable to obtain the EC50 value to compare with the concentration range used in the paper to confirm it is physiologically relevant concentration.

Responses:

We agree with the reviewer that the GABA application time may have been shorter than the bar in Fig .1a of the first revision of the manuscript indicated. This was partly because of the technical limitations of our drug application system. With GABA puffing in the perfusion solution, it was difficult to define the exact endpoint of GABA application in the first revision of the manuscript.

In response to the reviewer's comment, we have performed a new set of experiments using the Y-tube method (Murase K et al., *Neurosci Lett.* (1989) 103(1):56-63), and revised Fig. 1a to reflect the new results. The current traces now show that GABA-induced currents are activated only during GABA application periods (Fig. 1a of the second revision of the manuscript).

As the Reviewer suggested, we investigated and compared GABA-induced currents at different GABA concentrations in BMDMs. The results are summarized in Fig. 1a of the second revision of the manuscript. GABA induced inward currents at concentrations higher than 100 μ M, in a concentration-dependent manner, in the macrophages. These results are in accordance with our other results suggesting that at hundred micromolar concentrations, GABA facilitated autophagy in bacteria-infected macrophages and animals *in vivo*. The text has been revised accordingly, with the correct references added (Results, Page 5, lines 94-102; Materials and methods, Page 24, lines 571-578), as follows:

Results; Page 5, lines 94-102)

In the whole-cell patch-clamp mode, fast application of GABA to BMDMs using the Y-tube method evoked transient inward currents in a concentration-dependent manner (Fig. 1a). In the presence of bicuculline (BIC), a GABA_AR antagonist, GABA failed to cause an inward current ($n = 5$), suggesting that GABA predominantly evoked currents by binding to GABA_ARs. GABA-induced inward currents showed minimal desensitization during GABA application. Thus, we investigated whether GABA induced tonic current, which was defined as a holding current shift by the GABA_AR antagonist BIC (50 μ M), in the presence of GABA (Supplementary Fig. 1a).

Materials and Methods; Page 24, lines 571-578)

Patch pipettes were filled with a high-Cl⁻ solution with the following composition (in mM): 140 KCl, 10 HEPES, 5 Mg²⁺ATP, 0.9 MgCl₂, and 10 EGTA. Y-tube method⁶⁴ was used for rapid and focal GABA application onto macrophages in the extracellular recording solution (in mM): 126 NaCl, 2.5 KCl, 1 MgCl₂, 1.8 CaCl₂, 26 NaHCO₃, 1.25 NaH₂PO₄, pH 7.4, saturated with 95% O₂–5% CO₂ and 297 mOsm. Whole-cell currents were acquired and analyzed with pClamp 9.0 (Molecular Devices). GABA-induced current was defined as the difference between the holding current before and after the application of GABA.

We further discussed the physiological and pharmacological significance of GABA_AR currents at different GABA concentrations (Page 19, lines 447-454). (See also comment #2 and #4).

2. page 5, line 105, do the authors mean the slow perfusion (3ml/min) selectively activate tonic current but not evoked current instead of concentration-dependent? Normally GABA tonic currents are activated by high-affinity GABA_AR with low GABA concentration, while synaptic-like evoked GABA currents are activated by low-affinity GABA_AR. Here the authors showed 300 μ M GABA activated both synaptic-like and tonic current by different flow rate. It requires further clarification. In addition, fast-application was mentioned in the methods.

Responses:

Thank you for this insightful comment. We apologize for our mistake in the interpretation of the “synaptic-like” current in the first revision. For the first revision of our manuscript, we adopted the pressure-driven GABA puffing system for fast application of GABA to BMDMs,

in addition to the GABA application with perfusion solution (~3ml/min) used in the original manuscript. However, during the second revision, we found that the “synaptic-like” current is dependent on pressure for GABA puffing, and was not dependent on the GABA concentration. Using the Y-tube system, we confirmed that in BMDMs, only “tonic currents”, and not “synaptic-like” transient currents, could be detected. The representative results of long-lasting tonic currents in BMDMs in revised Fig. 1a was shown in more than 15 independent experiments. A detailed description of the Y-tube system for fast application has been included, with appropriate references, in the Materials and Methods section.

Fast and direct application of GABA to BMDMs using the Y-tube method induced inward currents in a concentration-dependent manner, which showed minimal desensitization under the fast GABA application condition (Fig. 1a of the second revision of the manuscript).

The results showed that, due to the minimal desensitization, the amplitudes of GABA-induced inward currents resulting from the fast GABA application were not significantly different from the tonic currents activated by GABA in the perfusion solution. Tonic current was defined as the holding current shift induced by the GABA_AR antagonist, bicuculline (50 μ M), in the presence of GABA. These results are now included in Supplementary Fig. 1a in the second revision of the manuscript. The text has been also revised accordingly (Page 5, lines 94-114 of the second revision of the manuscript).

As the reviewer suggested, we examined whether GABA activates high-affinity GABA_ARs to generate tonic currents in macrophages. The results have been included in the second revision of the manuscript (Supplementary Fig. 1b). The text has also been revised accordingly (Page 5, lines 102-114). In the presence of GABA, a preferential GABA_AR δ subunit agonist generated a tonic GABA_A current in a concentration-dependent manner, suggesting that BMDMs express GABA_ARs containing the δ subunit, thus generating a tonic current.

These data and comments have been included in Results (page 5, lines 102-114), as follows:

“GABA generated tonic current in a concentration-dependent manner, which had similar amplitudes to those of the inward currents during the fast GABA application ($P > 0.5$ at each GABA concentration); this suggests that GABA mainly evoked tonic currents in BMDMs. GABA_AR pentamers, assembled from 19 subunits, vary in subunit combinations, which diversifies GABA_AR functions. For example, the δ -subunit confers a slower desensitization on GABA_ARs, making them ideal candidates to generate tonic

GABA_A current in the brain²⁰. To determine whether this is the case in peripheral macrophages, we used 4,5,6,7-tetrahydroisothiazolo-[5,4-c]pyridin-3-ol (THIP), which preferentially activates the δ over the γ_2 subunit-containing GABA_ARs²¹⁻²³ (Supplementary Fig. 1b). THIP (1–30 μ M) generated tonic GABA_A current in a concentration-dependent manner in the presence of GABA (100 μ M), although THIP alone failed to significantly alter the holding current of BMDMs (data not shown). These results suggest that BMDMs express functional GABA_ARs that respond by generating tonic currents when exposed to GABA.”

Overall, our new results are in agreement with our original hypothesis that BMDMs express functional GABA_ARs, especially the δ subunit-containing receptors. However, given that the δ subunit confers high GABA sensitivity to GABA_ARs in the brain (Farrant M et al., Nat Rev Neurosci. (2005) 6(3): 215-29), our results showing that GABA activates tonic current at hundred micromolar concentrations suggest that a specific type of GABA_ARs are functionally active in BMDMs. The apparent discrepancy may be because GABA_AR subunits and/or their assembly in BMDMs are independent from GABA_AR pentamer assembly patterns in the brain. Future studies are warranted to characterize the exact assembly of GABA_ARs in peripheral macrophages.

These data and comments have been included in Discussion (page 15-16, lines 355-361), as follows:

“Our data extend these previous findings by confirming the ability of 11 GABA_AR subunit transcripts and functional receptors containing the δ subunits to generate tonic currents in resting BMDMs. Given that the δ subunit confers high GABA sensitivity to GABA_ARs in the brain²⁰, our results demonstrating that GABA in the order of a few hundred μ M activated tonic current suggest that a specific type of GABA_ARs are functionally active in BMDMs. Future studies are warranted to characterize the exact assembly of GABA_ARs in peripheral macrophages.”

3. page 5, line 117. It has been shown that lung airway epithelial cells also secrete GABA. The author needs to clarify where in the lung the image was taken and co-labeling to confirm the GABA-stained cells are macrophages (fig 1d).

Responses:

We further examined the major cellular sources of GABA in the lung tissues. Lung airway epithelial cells, as well as macrophages, were found to be the source of GABA (see

Supplementary Fig. 2). GABA expression was highly co-localized with the macrophage marker F4/80, as well as the alveolar type II epithelial (AE II) cell-specific marker ABCA3 in the lung tissues of mice (Supplementary Fig. 2). Therefore, both lung AE II cells and macrophages in mouse lung express endogenous GABA.

These comments have been included in the Results section (page 6, lines 121-124).

“GABA expression was highly colocalized with the macrophage marker F4/80, as well as the alveolar type II epithelial (AE II) cell-specific marker ABCA3 in murine lung tissues (Supplementary Fig. 2), suggesting that endogenous GABA is expressed by both lung AE II cells and macrophages in mouse lungs.”

4. page 6, line 119. The results showed the decrease of serum GABA in Mtb-infected mice (1f). The GABA concentration range is around 1.8-2.2 uM. How does this concentration relate to the GABA concentration used later in in vitro experiments (e.g. 100 uM)? This issue is very important because higher GABA concentration will desensitize some GABAAR and leads to stop of GABA signaling.

Responses:

The Reviewer asked us about the physiological relevance of GABA to our study. In Figs. 1–6, we aimed to characterize the effects of exogenous GABA on the activation of the host defense against intracellular bacterial infection. One of the major purposes of our study was to understand the role of GABAergic function in the development of several potential autophagy-targeting agents, which could serve as host-directed therapeutics to overcome drug resistance issues in a variety of infectious diseases, including TB⁵³⁻⁵⁵. (In Discussion, p. 19, lines 454-460). In our system, GABA treatment was beneficial for enhancement of the antimicrobial host defense against intracellular bacterial infections.

Regarding the physiological relevance of GABA in our study, please refer to the following: 1) Fig. 7; 2) the exogenous GABA concentration used *in vitro/in vivo* in previous studies; 3) behavioral tests showing that GABAAR signaling is not attenuated; and 4) the subunit composition of macrophage GABA_ARs containing the δ subunit. Detailed explanations about these four points are provided below:

1) Fig. 7

In Fig. 7, all *in vivo/in vitro* experiments were performed without GABA treatment *per se*. When the macrophages were treated with GABA_AR antagonists, the intracellular or *in vivo*

bacterial loads were significantly increased. These data strongly suggest that the basal levels of GABA and functional GABA_AR signaling are critically involved in the appropriate induction of host defenses against an intracellular bacterial infection.

2) The exogenous GABA concentration used in previous *in vivo/in vitro* studies did not attenuate GABAR signaling. The GABA concentrations used in this study (100 µM to 1 mM) were based on numerous previous reports (see Table 1 below for review purposes only) that investigated the function of GABA in the regulation of peripheral immune responses.

Table 1 for the review purpose only. GABA dose used *in vivo/in vitro* in previous literatures

In vivo, GABA administration			
	GABA	Route	Reference
Human	100mg	Oral	Abdou, A.M., et al., Biofactors. 2006;26(3):201-8
	800 mg	Oral	Steenbergen, L., et al., Sci Rep. 2015 Jul 31;5:12770.
	0.8g/kg	Oral	D. B. Tower, in "Inhibition in the Nervous System and Gamma-Aminobutyric Acid," p. 562-578. Pergamon Press, Oxford, 1960
Rat	100 mg/100 g body weight	Oral	Abdou, A.M., et al., Biofactors. 2006;26(3):201-8
	100 or 500 mg/kg	Oral	Kim, H.Y., et al., Food Chem Toxicol. 2004 Dec;42(12):2009-14.
	10 mg/kg, 100-500 mg/kg	intraparitoneal injection	Borycz, J., et al., J Physiol Pharmacol. 1992 Sep;43(3):259-69.
Mice	10 mg/kg	intraparitoneal injection	http://digitalcommons.cwu.edu/source/2014/posters/151/
	2-2000 mg/kg (mainly 200mg/kg)	intraparitoneal injection	Carmans S et al., J Neuroimmunol. 2013 Feb 15;255(1-2):45-53.
	25-2000 mg/kg (mainly 100 mg/kg)	intraparitoneal injection	Biswas, B., et al., Psychopharmacology (Berl). 1978 Sep 15;59(1):91-4.
	0.04-40 mg/kg	intraparitoneal injection	Ren W et al., Front Immunol. 2017 Jan 16;7:685.

In vitro, GABA treatment

	Cell type	Concentration	Reference
	Peritoneal macrophage	GABA 0.1-1 mM	Carmans S et al., J Neuroimmunol. 2013 Feb 15;255(1-2):45-53.
	Splenic mononuclear cells	GABA 0.1-3 mM	Tian J et al., J Immunol. 2004 Oct 15;173(8):5298-304.
Mice	Dendritic cells	GABA 0.5 μ M, Muscimol 300 μ M, Bicuculline 50 μ M	Fuks, J.M. et al., PLoS Pathog. 2012;8(12):e1003051.
	Peritoneal macrophages	Muscimol 10–100 μ M	Bhat R et al., Proc Natl Acad Sci U S A. 2010 Feb 9;107(6):2580-5.
	Dendritic cells	GABA 1 μ M or 1 mM	Fuks, J.M. et al., PLoS Pathog. 2012;8(12):e1003051.
Human	Dendritic cells	GABA 1 μ M or 1 mM	Fuks, J.M. et al., PLoS Pathog. 2012;8(12):e1003051.

3) Behavioral tests

Finally, we examined whether the maximal doses of GABA used in our study led to any undesirable effects, i.e., desensitization of some GABA_AR that would in turn lead to inhibition of GABA_AR signaling.

Figure 1 shows that (for review purposes only) GABA injection (200 mg/kg, i.p., q.d.) caused no significant behavioral changes, as determined by the open field test (Fig. 1a, b for review purposes only), the rotarod test (Fig. 1c for review purposes only), and seizure sensitivity to electroshock (Fig. 1d for review purposes only). This interpretation is consistent with results showing that a GABA injection (200–400 mg/kg, i.p., q.d.) did not affect brain GABA concentrations, while a higher concentration of GABA (600 mg/kg) increased brain GABA (Shyamaladevi N et al., Brain Res Bull. (2002) 57(2):231-236).

Figure 1. For review purposes only. Behavior tests of GABA-treated mice *in vivo*. (a) Rotarod testing. Time to fall during trials 1–4 of the rotarod test, 6 rpm, max time 300 s. Mice were treated with PBS ($n = 5$) or GABA ($n = 5$) and monitored for 17–21 days (the conditions are shown below). (b and c) Horizontal locomotor activity was measured using the open field test for 60 min. (b) Representative traces of mouse movement during the open field test. (c) There were no differences in total distance covered by mice between PBS ($n = 8$) and GABA ($n = 8$) treatment conditions at 18–21 days (the conditions are shown). (d) Seizure sensitivity to electroshock was compared between mice treated with PBS ($n = 5$) or GABA ($n = 5$) at 21 and 24 days (conditions are shown below).

Combined with the fact that GABA_AR antagonists themselves significantly increased the intracellular and *in vivo* bacterial loads (Fig. 7), these additional data support the notion that peripheral GABA administration does not induce desensitization of GABA_AR to inhibit GABA signaling. The authors politely argue against the concern that the GABA concentrations used in the current study may have affected GABA_AR signaling.

These comments have been included in the Discussion (Page 19, lines 447-454).

“Because the experiments using GABA_AR antagonists were performed in the absence of GABA treatment *per se*, the basal levels of GABA and functional GABA_AR signaling are critical for appropriate induction of host defenses against intracellular bacterial infections. Additionally, GABA injection (200 mg/kg) caused no significant behavioral

changes, as determined by the open field test, the rotarod test, and seizure sensitivity to electroshock (data not shown), indicating that the maximal doses of GABA used in our study did not have any undesirable effects on the neurological system or GABA_AR signaling.”

4) Subunit composition of macrophage GABA_ARs containing the δ subunit.

Combined with the fact that GABA_ARs containing the δ subunit show minimal desensitization when exposed to GABA (Farrant M et al., *Nat Rev Neurosci.* (2005) 6(3): 215-29), it is noteworthy that GABA induced long-lasting tonic currents, as mediated by GABA_ARs containing the δ subunit in BMDMs (Supplementary Fig. 1b of the second revision of the manuscript). The subunit composition of GABA_ARs could minimize receptor desensitization, even at high GABA concentrations, in macrophages. This idea is supported by the fact that THIP, a GABA_AR δ agonist subunit, activated tonic GABA_A current in the presence of 100 μ M GABA in BMDMs.

These data and comments have been included in Results (page 5, lines 102-114), as follows:

“GABA generated tonic current in a concentration-dependent manner, These results suggest that BMDMs express functional GABA_ARs that respond by generating tonic currents when exposed to GABA.”

5. page 6, line 13, GAD65/67 was increased in BMDM upon Mtb and BCG infection. Does it contradict to the decrease of GABA in BMDM shown in the fig. 1b and line 116?

Responses:

The GAD65/67 expression in macrophages was significantly increased after 18 h of Mtb or BCG infection. This may be due to feedback amplification in response to the intracellular GABA decrease in macrophages. Although we did not examine the time course of GABA synthesis in macrophages, we speculate that the increased GAD65/67 expression contributes to restore endogenous GABA during the later stages of infection.

These comments have been included in Discussion (page 16; lines 363-367), as follows:

“Interestingly, the protein expression of GAD65/67 in macrophages increased following infection. There is speculation that the activities of GABA-synthesizing

enzymes in macrophages can be increased by feedback activation during infection. Therefore, we examined the effects of exogenous GABA in the activation of host defense against intracellular bacterial infection.”

6. page 6, line 129-130. How could the authors claim these five subunits have relatively higher expression? Have all primer efficiency been tested?

Responses:

Although qRT-PCR is a powerful tool for the analysis of gene expression, we agree with your concerns that the results of relative quantification of qRT-PCR analysis could be considerably influenced by several factors, as previously reported (Regier N et al., BMC Mol Biol. (2010) Aug 11; 11:57). We agree with your opinion that we should be careful when describing the relative expression.

Thus we have corrected the sentence in Results (page 6, lines 136-137), as follows:

We then selected five subunits (*Gabra3*, *Gabra4*, *Gabrb3*, *Gabrd*, and *Gabrq*) that were detectable in BMDMs, and examined their mRNA levels before and after Mtb infection.

Primer efficiencies were evaluated using a standard curve, generated by real-time PCR using a 10-fold dilution series over at least four dilution points; these were measured in triplicate. Cycles versus cDNA concentration input were plotted to calculate the regression slope. The reaction efficiency (E) for each gene was calculated using the equation $E = 10^{1/\text{slope}} - 1$ (Presslauer C et al., PLoS ONE. (2014) 9(12): e114209). The reaction efficiencies of the primers used in our study were between 90–110%, as shown in Table 2 (for review purposes only).

Table 2 for the review purpose only. The primer efficiencies of 5 murine and 3 human GABA_AR subunits in this study.

Human Genes	E %
-------------	-----

Mouse Genes	E %
Gapdh	96.2
Gabra3	107
Gabrb3	107.7
Gabra4	91.4
Gabrq	103.4
Gabrd	107.7

GAPDH	101.1
GABRA1	95.4
GABRR2	96.7

7. page 7, line 151-159. Could be the activation of GABAAR expressed in other cell types in the lung rather than macrophage contribute the difference of Tnf and Il6 expression (lung tissue vs. BMDM)? It is more convincing to measure Tnf and Il6 protein level by e.g. ELISA.

Responses:

To examine the protein levels in BMDMs, we performed the ELISA analyses using the supernatant from Mtb-infected BMDMs with or without GABA treatment. We found that TNF and IL-6 levels were decreased in BMDMs after GABA treatment, and the levels were similar to those found in lung tissues during Mtb infection. These data suggest that GABA treatment led to an inhibition of proinflammatory cytokine generation *in vivo* and *in vitro* during Mtb infection.

We have corrected the description in Results (page 7-8, lines 167-170), as follows:

“In BMDMs, GABA treatment led to a significant reduction in TNF α and IL-6 protein expression in response to Mtb infection (Supplementary Fig. 7c, d). Taken together, these results indicate that GABA treatment inhibited proinflammatory cytokine production *in vivo* and *in vitro* during Mtb infection.”

However, we did not perform ELISA analysis on the lung lysates because we were unable to remove the samples from the A-BSL3 facility due to regulations.

8. page 8, line 185. Are these genes significantly up-regulated? If so, please describe the statistical method used in detail. The expression of *Tnf* and *Il6* was not detected in RNAseq but detected in qPCR with only 30 PCR cycles. The authors need to clarify this.

Responses:

Fig. 3a shows the heatmap analysis of the expression of numerous genes involved in autophagy. The number of biological replicates of RNAseq analysis was insufficiently high (duplicate) for statistical tests to be applied. Thus, we performed qRT-PCR analysis on more samples ($n = 9$) to validate the RNAseq analysis.

We have corrected the sentence, as follows (page 9, line 195-196):

“Numerous genes involved in autophagy were substantially upregulated ...”

In Supplementary Fig. 7, we show that the mRNA expression of inflammatory cytokines *Tnf* and *Il6* were increased by *Mtb* infection, and that the *Tnf* mRNA levels were downregulated by GABA treatment in *Mtb*-infected BMDMs. GABA *per se* did not induce proinflammatory cytokine expression, as shown in Supplementary Fig. 7c and d. For RNAseq analysis (Fig. 3a), we examined the global mRNA profile of BMDMs after GABA treatment alone. Therefore, these are not controversial results.

9. page 8, line 191-192. How does qPCR correlate with the above-mentioned RNAseq data?

Responses:

As we stated above, we performed qRT-PCR analysis to validate the RNAseq data. Most genes, including *Gabarap1*, *Gabarap2*, *Gabarap*, *Lamp1*, *Map1lc3b*, *Ambra1*, *Atg4c*, and *Ulk1*, which showed a substantial increase by RNAseq analysis, were also found to be significantly upregulated by qRT-PCR analysis. However, the mRNA levels of a few genes, including *Becn1* and *Atg7*, were not considerably changed in the qRT-PCR analysis after 6 h of GABA treatment.

According to the validation study using qRT-PCR analysis, we selected *Gabarap1* for further experiments.

10. page 11, line 260. How does GABA signaling relate to AMPK? Can the author pre-block GABAAR with e.g PTX and show the decrease of AMPK activation?

Responses:

We performed the experiment using bicuculline instead of PTX, because bicuculline is the benchmark antagonist for GABA_AR (Johnston GA et al., Br J Pharmacol. (2013) 169(2): 328–336).

We have included the western blot analysis data showing that GABA-induced AMPK activation was significantly inhibited in BMDMs in the presence of bicuculline in a dose-dependent manner. These data suggest that GABA_AR blockade leads to an inhibition of GABA-triggered AMPK activation in BMDMs.

These data have been included in Results (page 11, lines 267-270), as follows:

“In addition, GABA-induced AMPK activation was significantly inhibited in BMDMs in the presence of either BIC or BAPTA-AM in a dose-dependent manner (Supplementary Fig. 12a, b), suggesting the contribution of GABA_AR or [Ca²⁺]_i increase in GABA-mediated activation of AMPK.”

11. page 11, line 273. The increase of gabarapl2 is not convincing in fig 6a.

Responses:

We apologize for our mistake in the analysis of *Gabarapl2*. There was no significant increase in *Gabarapl2* mRNA expression by GABAergic activation in Mtb-infected BMDMs (Fig. 6a).

We have corrected the Figure 6a and the description in the Results (page 12, lines 283-284).

.....the levels of mRNAs encoding *Gabarapl1*, but not *Gabarapl2*, in Mtb-infected BMDMs (Fig. 6a).

12. page 11, lin 274. Why not use primary cells in Gabarapl1-overexpression experiment? The expression of GABAAR in cell line may differ from primary cells as previously shown.

Responses:

We agree with the reviewer's point that primary cells would be better to interpret the function of *Gabarapl1* than RAW264.7 cells. However, the *Gabarapl1* adenovirus system for overexpression studies in primary cells was not available in our lab. In addition, we showed another two experimental approaches for functional investigation of *Gabarapl1* in primary cells using *Gabarapl1* knockdown (Fig. 6g, e, f; Supplementary Fig. 14c and d) and *Gabarapl1*-null BMDMs (Fig. 6h, Supplementary Fig. 14a and b).

We also found that RAW264.7 cells showed considerable expression of five subunits (*Gabra2*, *Gabra3*, *Gabra5*, *Gabrb3*, and *Gabrd*) of GABA_AR. The mRNA expression profile of GABA_AR subunits in RAW264.7 cells is shown in Supplementary Fig. 13 and the Results section (page 12, lines 287-289), as follows:

"In RAW264.7 cells, we found that the mRNAs of several GABA_AR subunits (*Gabra2*, *Gabra3*, *Gabra5*, *Gabrb3*, and *Gabrd*) were present, as measured by qRT-PCR analysis (Supplementary Fig. 13)."

13. page 12-13. Can the author show the application of GABA or GABA-agonist could decrease the bacterial burden in zebrafish and flies?

Responses:

We have included data showing that GABA treatment led to a decrease in *M. marinum* loads in zebrafish and flies. As shown in Supplementary Fig. 6, GABA treatment in zebrafishes led to a significant decrease in mycobacterial burden. These comments have been included in Supplementary Fig. 6 and the Results section (page 7, lines 154-157), as follows:

"We then used *M. marinum* to establish a zebrafish embryo model of infection. Infected embryos were exposed to GABA and bacterial counts were measured. As shown in Supplementary Fig. 6, the bacterial burdens were significantly lower in zebrafish embryos grown with GABA."

Although we have preliminary data showing that GABAergic activation decreased *in vivo* bacterial loads (Figure 2 for review purposes only), we did not include these data in this manuscript and would like to keep them for a future paper.

Figure 2. For review purposes only. GABAergic activation decreases the bacterial loads in flies. WT flies were injected with 500 CFU *M. marinum* and then maintained with or without muscimol (Mus; 100 μ M) or isoguvacine hydrochloride (Iso; 100 μ M). After 3 days, each group ($n = 10$ per group) of flies was homogenized, harvested, and quantified using a CFU assay.

14. page 15, line 354. Not all combination of GABAAR subunits can form functional receptor ion channels, so it is possible to apply specific compounds to study the combination of e.g. $\alpha 4\beta 3\delta$ in this study.

Responses:

GABA_AR pentamers assembled from 19 subunits vary based on the subunit combination, which lead to diverse GABA_AR functions. For example, GABA_ARs with distinct subunit compositions localize differentially between synaptic and non-synaptic sites, to mediate phasic and tonic inhibition in the brain, respectively. Despite a multitude of theoretical permutations, limited subunit combinations have been identified in the brain.

We agree with the reviewer that it is important to determine the subunit composition of GABA_AR in macrophages. As the reviewer suggested, we showed that THIP, a preferential GABA_ARs δ subunit agonist, activated tonic GABA_A current in the presence of GABA in BMDMs (Supplementary Fig. 1b of the second revision of the manuscript), suggesting that GABA_ARs containing the δ subunit are functionally expressed and activated when exposed to GABA.

These data and comments have been included in Results (page 5, lines 102-114), as follows:

“GABA generated tonic current in a concentration-dependent manner, which had similar amplitudes to those of the inward currents during the fast GABA application ($P > 0.5$ at each GABA concentration); this suggests that GABA mainly evoked tonic currents in BMDMs. GABA_AR pentamers, assembled from 19 subunits, vary in subunit combinations, which diversifies GABA_AR functions. For example, the δ -subunit confers a slower desensitization on GABA_ARs, making them ideal candidates to generate tonic GABA_A current in the brain²⁰. To determine whether this is the case in peripheral macrophages, we used 4,5,6,7-tetrahydroisothiazolo-[5,4-c]pyridin-3-ol (THIP), which preferentially activates the δ over the γ_2 subunit-containing GABA_ARs²¹⁻²³ (Supplementary Fig. 1b). THIP (1–30 μ M) generated tonic GABA_A current in a concentration-dependent manner in the presence of GABA (100 μ M), although THIP alone failed to significantly alter the holding current of BMDMs (data not shown). These results suggest that BMDMs express functional GABA_ARs that respond by generating tonic currents when exposed to GABA.”

15. page 15, line 359. Neutrophil also expresses GABAR receptors, so GABA could have direct effect on neutrophil infiltration.

Responses:

We were asked by the reviewer to show the effect of GABA on neutrophil infiltration, and thus modulation of immunopathology/inflammation during Mtb infection. Therefore, we further examined the effects of GABA on neutrophil infiltration in the infected lung tissues during Mtb infection.

We found that polymorphonuclear neutrophil (PMN) infiltration in Mtb-infected lungs was significantly decreased by GABA treatment (Supplementary Fig. 5d). Thus, GABA treatment may contribute to amelioration of the immune response and neutrophil infiltration during Mtb and BCG infection.

These comments have been included in Results (page 7, lines 151-154), as follows:

“Additionally, the number of granulomatous lung lesions and neutrophil infiltration decreased significantly (compared with controls) following GABA treatment after intravenous (i.v.) Mtb (Fig. 2d, for granuloma; Supplementary Fig. 5d, for neutrophil infiltration) or i.v. BCG challenge (Supplementary Fig. 5b).”

We agree with the comment that GABA-mediated inhibition of neutrophil infiltration may be due to the direct effect of GABA on neutrophils, since neutrophils also express GABA_AR. However, the issue of whether neutrophils respond to GABA and affect host defense is beyond the scope of this paper. Therefore, we thank the reviewer for their suggestion and will consider this for a future study.

We have included the points in Discussion (Page 16, lines 380-385), as follows:

“We found that GABA treatment ameliorated inflammatory cytokine production and neutrophil infiltration in the infected sites (lungs) during Mtb infection. Because neutrophils also express GABA_ARs⁴³, GABA-mediated inhibition of neutrophil infiltration may be due to direct effects of GABA on neutrophils at the site of infection. Future studies are needed to clarify whether neutrophils directly respond to GABA and contribute to host defense during infection.”

16. page 23, line 600-606. Has semiquantitative RT-PCR been used in the study? If not, please remove it from the subtitle. Which chemistry (SYBR or probe) was used for qPCR? Why was GAPDH used as the reference gene here but b-actin used later (supplementary material)? Have the author tested the stability of reference gene expression under the experiment conditions? Are all primers validated with efficiency? Do they cross exon-exon junction or on different exons? The detailed information for primers should be shown in a table. Have the RNA been treated with DNase I to remove the potential genomic DNA contamination?

Responses:

We performed semi-quantitative RT-PCR analysis and the results are shown in Figs. 5h and 6e. We have corrected the subtitle and description (page 26, line 639).

A Rotor Gene SYBR Green PCR kit (Qiagen; 204074) was used for qRT-PCR analysis in this study. We tested two reference genes, *Gapdh* and *Actb*, in our experiments. We did not find any significant differences in the C_q values of the different treatment combinations between the two reference genes.

All primers were validated for efficiency, as stated in the response to comment #6. We typically designed primers (either the forward or reverse primer) that cross an exon-exon junction.

Primer information is summarized in **Supplementary Table 1**.

RNA samples were treated with DNase I to remove potential genomic DNA contamination.

17. In general, most of the statistical analyses are applied to the data from duplicates or triplicates of two experiments. The results will be more convincing if 3 or 4 independent experiments.

Responses:

We apologize for missing out the number of independent experiments. We performed at least three independent experiments for most figures, which yielded one representative data set. Therefore, we have corrected the figure legends to include the number of independent experiments conducted.

However, for the *in vivo* Mtb infection examination, we only performed two independent experiments (Figs. 1f, 2a, 7a Mtb) due to the limitations of the ABSL3 facility. In addition, we have replaced some data by combining the results of three or four experiments (Figs. 2d, 2h, 7a BCG; and Supplementary Fig. 15a).

18. The image panel layout should be organized so the reader can easily follow.

Responses:

We have organized the image panel layout following the order in each figure.

19. Please be consistent with the labels of control, U, PBS, sc in all figures. Did the authors use the same solvent (PBS) or sometimes with DMSO? It should be clearly stated, especially for PTX, GABA, and other GABA agonist and antagonist?

Responses:

In our experiments, we kept equimolar solvent concentrations between control and test samples, even at very low concentrations. When we used several agonists/antagonists in the same experiment, we confirmed that each solvent control did not induce any undesirable cellular or cytotoxic effects and showed similar results to those (PBS control) for GABA.

Therefore, the SC data in most figures were for GABA (0.1 % PBS). In some figures (2e, 2f, 3d-j, 4a-d, 4i, 5a-d, 5g-k, 6e-h, 7e-h), other SC data [muscimol, 0.05 μ M HCl; picrotoxin, 0.05% EtOH; Bafilomycin A and BAPTA-AM, 0.1% DMSO; isoguvacine hydrochloride, PTZ, and Bicuculline, 0.1% D.W.] were used as a control. Detailed solvent information is described in the respective figure legends.

Furthermore, we have corrected all 'control, U, PBS, and sc' labels to U (untreated), UI (uninfected), PBS (for GABA), or SC (for GABA or other reagents; clarified in each experiment)' throughout the paper.

20. Fig 1 d and e. The figures showed GABA staining in uninfected vs Mtb infected, but the figure legend in lin 1004 wrote "treated with PBS or GABA (i.p. 200 mg/kg)". Is it strange?

Responses:

We apologize for our mistake and have corrected the description in the Fig. 1 legend (page 39, lines 950-951), as follows:

d-f Mice were infected i.v. with Mtb (1×10^6 CFU), and monitored at 14 dpi.

21. line1013, is the expression supposed to normalize to reference gene rather than "infected BMDM as the calibrator"?

Responses:

We apologize for our mistake and have corrected the description (line 958-959), as follows:

....and normalized to the reference gene *Gapdh*.

22. line 1015, fig 1k. Other cell types in the lung also express GABAAR: How do the authors differentiate the expression from macrophage?

Responses:

We agree with the reviewer. It is possible that other cell types in the lung also express various GABA_AR subunits. However, staining lung tissues with various combinations of Abs (not only for the individual subunit of GABA_ARs, but also for several cell types) would require extensive experimentation, as would or sorting the cells from lung tissues. Therefore, we are asking to the reviewer to consider this as a possibility for a future paper.

23. Is figure 2b lung the same as fig S3b?

Responses:

The experimental procedures differ between Fig. 2b and Supplementary Fig. 5a. In addition, the routes were intravenous and intranasal for Fig. 2b and Supplementary Fig. 5a, respectively.

These comments have been included in Fig. 2b legend (page 41, line 972), as follows:

“**b** Mice ($n = 8$ per group) were infected i.v. with BCG (1×10^7 CFU),..”

and in Supplementary Fig. 5a legend (Supple text page 10, lines 101-103), as follows:

“Mice ($n = 8$ per group) were infected i.n. with BCG (2×10^6 CFU)....”

We have corrected Supplementary Fig. 5a to reflect more representative data in the revised manuscript.

24. line 1034-1036. Do human MDMs have similar GABAAR expression pattern as in mouse BMDMs?

Responses:

We performed the qRT-PCR analysis for human primary MDMs for 19 GABA_AR subunits. As shown in Figure 3 for review purposes only, the expression levels of three subunits (*GABRA1*, *GABRB1*, and *GABRR2*) were detectable, whereas the mRNA levels of the other subunits were undetectable in resting, unstimulated MDMs. However, we would like to show these data in a future paper, and have thus excluded them from this manuscript.

These comments have been included in Results (page 7, lines 161-162), as follows:

In human primary monocyte-derived macrophages (MDMs), we found 3 mRNAs (*GABRA1*, *GABRB1*, and *GABRR2*) were detectable by qRT-PCR analysis (data not shown).

Figure 3 for review purposes only. Expression of GABA_AR subunits in human MDMs.

In Human MDMs, three GABA_AR subunit mRNAs were determined using the $\Delta\Delta C_t$ method and normalized to the reference gene *GAPDH*.

25. line 1048, how many RNAseq experiments were performed? Does the scale bar in fig 3a indicate increase of 0.1,0.2 or 0.3-fold? It is bit confusing.

Responses:

The RNAseq data are representative of duplicate determinations. These comments have been included in the Fig. 3 legend (page 43, lines 993-994), as follows:

“a Heatmap analysis of the RNAseq data (representative of duplicate determinations).”

We apologize for our mistake when describing the scale bar. We have corrected this.

26. fig 6a. Is it supposed to compare the difference of mtb vs GABA+,mtb, mtb vs iso+mtb, mus vs mus+mtb rather than GABA vs. GABA+mtb....?

Responses:

We have compared the differences of each value as you suggested.

27. Supplementary line 92. Is GABA missing from the sentence?

Responses:

We have corrected the sentence with addition of GABA (i.p., 200 mg/kg) in Supplementary line 102.

28. Supplementary line 97-98, Why do different statistical analysis used for the same type of data (d vs a and b)? The label of control or GABA should be consistent in a,b and d.

Responses:

We have applied the same statistical analysis (Mann-Whitney U test) to the same type of data.

The label of control or GABA has been corrected such that it is consistent throughout the paper.

29. Fig. S9e. Have the authors shown the increase of Ca²⁺ leading to AMPK activation in the experiments?

Responses:

We have included the western blot analysis data showing that GABA-induced AMPK activation was significantly inhibited in BMDMs in the presence of BAPTA-AM in a dose-dependent manner. These data suggest that intracellular Ca²⁺ influx is required for GABA-triggered AMPK activation in BMDMs.

These data have been included in Results (page 11, lines 267-270), as follows:

In addition, GABA-induced AMPK activation was significantly inhibited in BMDMs in the presence of either BIC or BAPTA-AM in a dose-dependent manner (Supplementary Fig. 12a, b), suggesting the contribution of GABA_AR or [Ca²⁺]_i increase in GABA-mediated activation of AMPK.

Minor comments:

1. The paper used both macrophages from mouse and human. Please state it clearly in the results, e.g. page5, line 97.

Responses:

As you commented, we have included the exact species, for examples, in murine BMDMs (page 5, line 92), and in human monocyte-derived macrophages (MDMs) (page 7, line 159).

2. page 23, line 561. RNAseq is not microarray, so please change the subtitle.

Responses:

We have corrected (page 25, line 607), as you commented.

3. page 28, The section “mRNA expression” can be combined with previous qPCR section.

Responses:

We have corrected (page 26, line 639; Section “RNA extraction and RT-PCR analysis”), as you commented.

4. line 1027, please state GABA concentration.

Responses:

We have included the exact GABA concentration (200 mg/kg) in page 45, line 973, 975

REVIEWERS' COMMENTS:

Reviewer #4 (Remarks to the Author):

I would like to thank the authors for including additional data to address my comments. I only have some minor comments below on the revised version.

Response:

Thanks for your valuable comments. We submit a revised version of our manuscript and a point-by-point response to the reviewer's comments. Detailed responses are described below.

Fig S1b, representative trace for 3 uM THIP, is BIC application missing from this trace?

Responses:

We have corrected the line of BIC application in the revised Supplementary Figure 1b.

Fig S2 merged images, please indicate in the figure legend that DAPI stainings are also included in the merged images.

Responses:

We have included the DAPI staining in the figure legend in Supplementary Figure 2.

Fig S12 bar graph, do 100 Bic+ 100 GABA groups have 4 independent experiments? The error bars were not seen.

Responses:

We apologize for our mistake of missing error bars in the revised Supplementary Figure 15. The error bars have been included in Supplementary Figure 15 bar graph, for 100 Bic+ 100 GABA groups.

Fig 6a, the error bar of Mtb+Iso group (Gabarapl1) was cut off.

Responses:

We have corrected the error bar of Mtb+Iso group (Gabarapl1) in the revised Figure 6a.

The authors have detected only 3 GABAAR (A1, B1 and R2) in human primary MDMS, which is totally different from the results in mouse BMDMs. This may reflect the differential effect of GABA on human and mouse macrophages. The authors are advised to add some comments to address this issue.

Responses:

We have included the human MDM data in Supplementary Figure 8.

In addition, we have added the description in Discussion (page 14-15, lines 359-364), as follows:

In human primary MDMs, we detected only 3 GABA_AR (*GABRA1*, *GABRB1*, and *GABRR2*), which expression profiles were different from those found in murine BMDMs, suggesting the differential effects of GABA on human and murine peripheral macrophages. Future studies are warranted to characterize the exact assembly and function of GABA_ARs in peripheral macrophages from different species.